# Stab-SGD: Noise-Adaptivity in Smooth Optimization with Stability Ratios

**David A. R. Robin**
INRIA - ENS Paris
PSL Research University

**Killian Bakong**
INRIA - ENS Paris
PSL Research University

**Kevin Scaman**
INRIA - ENS Paris
PSL Research University

## Abstract

In the context of smooth stochastic optimization with first order methods, we introduce the stability ratio of gradient estimates, as a measure of local relative noise level, from zero for pure noise to one for negligible noise. We show that a schedule-free variant (Stab-SGD) of stochastic gradient descent obtained by just shrinking the learning rate by the stability ratio achieves real adaptivity to noise levels (i.e. without tuning hyperparameters to the gradient's variance), with all key properties of a good schedule-free algorithm: neither plateau nor explosion at intialization, and no saturation of the loss. We believe this theoretical development reveals the importance of estimating the local stability ratio in the construction of well-behaved (last-iterate) schedule-free algorithms, particularly when hyperparameter-tuning budgets are a small fraction of the total budget, since noise-adaptivity and cheaper horizon-free tuning are most crucial in this regime.

We consider the standard Machine Learning setup, where the task of learning a function $f : \mathbb{R}^q \to \mathbb{R}^k$ from samples $(X \in \mathbb{R}^q, Y \in \mathcal{Y}) \sim \mathcal{D}$ is decomposed into a parameterization $F : \mathbb{R}^d \times \mathbb{R}^q \to \mathbb{R}^k$ and a loss function $\ell : \mathcal{Y} \times \mathbb{R}^k \to \mathbb{R}$, with the aim to minimize $\mathbb{E}[\ell(Y, f(x))|X = x]$. A parameter $\theta \in \mathbb{R}^d$ yields a predicted function $f_\theta = F(\theta, -) : \mathbb{R}^q \to \mathbb{R}^k$, whose quality is evaluated according to $\mathcal{L}(\theta) = \mathbb{E}_{X,Y}\left[\ell(Y, F(\theta, X))\right]$ defining a loss function $\mathcal{L} : \mathbb{R}^d \to \mathbb{R}$ to be minimized.

Typical scenarios include least-squares regression $\ell(u, v) = \|u - v\|_2^2$ for $\mathcal{Y} = \mathbb{R}^k$, with functional optimum $x \mapsto \mathbb{E}[Y|X = x]$; and classification with cross-entropy $\ell(y, u) = -u_y + \log \sum_i \exp(u_i)$ for $\mathcal{Y} = [k]$. The success of deep learning has taken this long past the historically well-studied linear case of $d = q \times k$, with impressive empirical performance lacking a strong theoretical support.

Using small batches of data to estimate gradients is one of the keys used to scale up such settings, leading to stochastic iterative algorithms. This randomness induces failures of constant-step gradient descents, which saturate and fail to minimize the loss past a threshold (e.g. Wilson and Martinez [2001]). This leads to the use of schedulers to shrink the learning rate over time. Setting it too low slows down optimization, and too high recovers saturated losses, thus even more hyperparameters are added to define schedulers of varying decay rates such as $\eta_t = \eta_0 \cdot t^{-\alpha}$ for $\alpha \in [0, 1]$.

**Related works.** The elimination of such hyperparameters, by a theory-backed choice of algorithm, has naturally been an active study of research. Such tentatives includes the early "Adagrad" [Duchi et al., 2011] and "Adadelta" [Zeiler, 2012] adaptive algorithms, but also "AC-SA" [Lan, 2012, Sec 3.1] and its more recent variants such as "Schedule-free SGD" [Defazio et al., 2024]. One branch of this effort chose to model the loss $\mathcal{L}$ as Lipschitz, i.e. having bounded gradients, see for instance the "COCOB" [Orabona and Tommasi, 2017, Thm 1] and "D-Adapt" algorithms [Defazio and Mishchenko, 2023, Thm 3] with known Lipschitz constant. Despite the immediate incompatibility with the least-squares objective, this modeling choice is supported by the Lipschitz-continuity of the ReLU non-linearity $x \mapsto \max(0, x)$ which is not differentiable (and thus not smooth) at the origin.

39th Conference on Neural Information Processing Systems (NeurIPS 2025).

The Lipschitz-model, typically used with convexity of $\mathcal{L}$ in addition, does not produce guarantees for the last iterate, but for the average $\frac{1}{T}\sum_t x_t$ or ergodic average $\sum_t \eta_t x_t / \sum_t \eta_t$ of iterates [Garrigos and Gower, 2023, Thm 9.6 - 9.12]. On the contrary, there is growing evidence that such aggregation is not mandatory[1] (e.g. the same reference Orabona and Tommasi [2017, Algorithm 2] from the Lipschitz-model branch does not use averaging on neural network experiments), and possibly detrimental in non-convex cases [Zhou et al., 2020, Figure 4]. Other requirements such as bounded domain are also questionnable. A second branch of this research effort thus focuses on a smooth model of the loss $\mathcal{L}$, i.e. Lipschitz-continuous gradients, which yield good last-iterate predictions (see Garrigos and Gower [2023, Thm 4.3] for the deterministic case and Bach and Moulines [2011, Thm 4] for the stochastic case with power schedule). By continuous-differentiability, these losses have gradients converging to zero near the global minimum which naturally leads to smaller steps, contrary to Lipschitz losses. This lack of averaging is also supported, outside the convex case using Jensen's inequality, by the lack of guarantees on the loss of the average iterate, even if the averaged loss is controlled.

Although this smooth model does not immediately fit the ReLU-based networks, experiments with smooth non-linearities often match performance of ReLU networks [Clevert et al., 2016, Elfwing et al., 2018, Sitzmann et al., 2020]. Moreover, any continuously differentiable function is smooth on compact domains, which supports the idea that this model will also be a good description of training dynamics naturally constrained to a compact set, e.g. by a regularization.

**Contributions.** We introduce in Sec. 1 the stability ratio, as a measure of gradient stochasticity, and as a shrinkage of SGD learning rates to obtain an algorithm adaptive to noise levels, formalized in Sec. 2. We show that this ratio is computable from samples and give an estimator. We prove in Sec. 3 how this adaptively achieves the optimal last-iterate rates of SGD at various noise levels, without the need to tune the learning rate to the (unknown) noise level or training horizon. We validate these statements with experiments in convex and deep learning scenarios in Sec. 4.

## 1 Stability Ratio: ensuring (strict) expected loss decrease

In gradient descents with large amounts of noise, a common practice is to shrink the step-size, backed by the standard intuition that lower learning rates are required to converge to low loss values. To quantify how much lower, we define a measure of "relative" or "normalized" noise level (between zero and one), inspired by classical smooth stochastic analysis, and show shrinking by this quantity achieved the desired adaptive result. For a random variable $X \in \mathbb{R}^d$ (not identically zero) with $0 < \mathbb{E}\left[\|X\|_2^2\right] < +\infty$, we denote as "Stability Ratio" the quantity $\text{Stab}(X) \in [0, 1]$ defined by

$$\text{Stab}(X) = \frac{\|\mathbb{E}[X]\|_2^2}{\mathbb{E}\left[\|X\|_2^2\right]}$$

Note, for $\mu = \mathbb{E}[X] \neq 0$, that $\mathbb{V}[X] = \sigma^2$ implies $\text{Stab}(X) = 1/(1 + \sigma^2/\|\mu\|_2^2)$, thus smaller variance leads to a stability ratio closer to 1. On the other hand, near-zero mean and non-negligible variance give stability ratios approaching zero: these are the estimates causing instabilities in the loss. The lower the stability ratio of the gradient, the lower the step-size must be taken to avoid instability.

For an SGD sequence $(\theta_t \in \mathbb{R}^d)$, using unbiased[2] stochastic gradient estimates $G_{t+1} \approx \nabla\mathcal{L}(\theta_t)$ to compute $\theta_{t+1} = \theta_t - \eta_t G_{t+1}$, the loss variation for a $\beta$-smooth function is at most

$$\mathcal{L}(\theta_{t+1}) - \mathcal{L}(\theta_t) \leq -\eta_t \cdot (\nabla\mathcal{L}(\theta_t) \cdot G_{t+1}) + \frac{\beta}{2}\eta_t^2 \|G_{t+1}\|_2^2$$

When $G_{t+1} = \nabla\mathcal{L}(\theta_t)$, this is minimized at $\eta_t = 1/\beta$, as in classical smooth deterministic analysis. In the stochastic case, taking the expectation and minimizing immediately gives $\eta_t = \text{Stab}(G_{t+1})/\beta$. Moreover, $\eta_t \leq \text{Stab}(G_{t+1})/\beta$ ensures that $\mathbb{E}_{G_{t+1}}[\mathcal{L}(\theta_{t+1})] - \mathcal{L}(\theta_t) \leq -\eta_t\|\nabla\mathcal{L}(\theta_t)\|_2^2/2$, and thus a decrease similar to that of gradient flow. Convergence is slowed down by a factor $\text{Stab}(G_{t+1})$, that is equal to 1 in the deterministic regime, and small in the high variance regime (where $\|\nabla\mathcal{L}(\theta_t)\|_2^2 \approx 0$ and $\mathbb{E}\left[\|G_{t+1}\|_2^2\right] \gg 1$). In what follows, we refer to SGD with such adaptive step-sizes as *Stab-SGD*, and discuss how to estimate this stability ratio in practice in Sec. 2.2.

---

[1] This claim is also supported for instance by the GPT3 training, which uses Adam without averaging [Brown et al., 2020, Appendix B p43], and the MuZero training, which uses a momentum version without averaging Schrittwieser et al. [2019] (see Ancillary file "pseudocode.py", L553).

[2] formally, satisfying $\mathbb{E}[G_{t+1} \mid \theta_t] = \nabla\mathcal{L}(\theta_t)$ with finite second moment $\mathbb{E}[\|G_{t+1}\|_2^2 \mid \theta_t] < +\infty$.

## 1.1 Adaptivity to noise level of stability-adjusted learning rates

Two typical regimes of SGD are depicted in Figure 1.1, with quadratic problems and injected additive gaussian noise $\varepsilon \sim \mathcal{N}(0, \sigma_0^2 I)$ for gradient estimates (varying $\sigma_0$), for a total variance of $\sigma^2 = \sigma_0^2 d$.

$$\mathcal{L} : x \in \mathbb{R}^d \mapsto \frac{1}{2} \sum_{i<d} \frac{1}{1+i} x_i^2, \qquad x^0 = (1)_{i \in [d]} \in \mathbb{R}^d, \quad d = 250 \tag{QSC}$$

$$\mathcal{L} : x \in \mathbb{R}^d \mapsto \frac{1}{2} \sum_{i<d} 2^{-i} x_i^2, \qquad x^0 = \left(2^{-i}\right)_{i \in [d]} \in \mathbb{R}^d, \quad d = 25 \tag{QWC}$$

Both losses are smooth with parameter $\beta = 1$. Problem QSC has $\mathcal{L}(x^0) \approx 3.05$, and is $\mu_0$-strongly convex with $\mu_0 = 1/250 = 4 \cdot 10^{-3}$. On the other hand, Problem QWC has $\|x^0 - x^\star\|_2^2 \approx 1.33$, $\mathcal{L}(x^0) \approx 0.5714$, and is $\mu_1$-strongly convex with $\mu_1 = 2^{-25} \approx 3 \cdot 10^{-8}$, which is too small to play a quantitative role in experiments, hence it is likely better described by weakly-convex smooth theory.

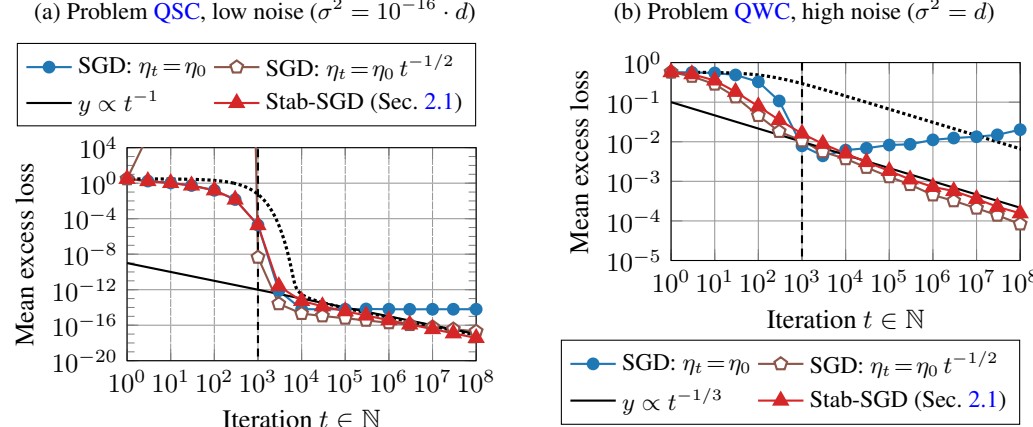

(a) Problem QSC, low noise ($\sigma^2 = 10^{-16} \cdot d$)    (b) Problem QWC, high noise ($\sigma^2 = d$)

Figure 1: Mean excess loss of SGD and variants. The hyperparameter of SGD (both constant and scheduled) is tuned by grid search at $10^3$ iterations (vertical dashed line). Bounds of Sec. 3 are presented with dotted lines. Details of all experimental protocols deferred to Sec. 4.

In (near-)deterministic settings, large step sizes are necessary, and decreasing too much gives slow asymptotic convergence (see Fig. 1a). Fast-decreasing schedulers emulating these large (near-constant) learning rates need huge initial learning rates, causing initial explosions which are prohibitive in deep learning. On the contrary, in more noisy settings (see Fig. 1b), shrinking the learning rate sufficiently is necessary, and constant learning rates trying to lower the saturation threshold will use much lower learning rates causing large initial plateaus. In both cases, the trajectory of Stab-SGD seems a more reasonable balance to strive for: no explosion, no initial plateau, no saturation.

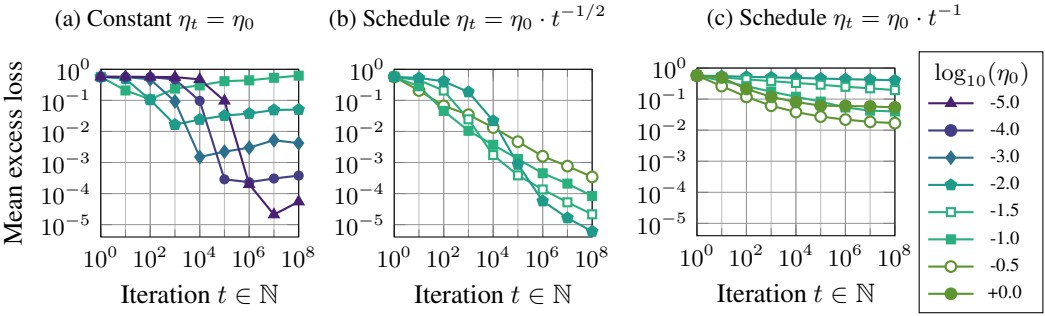

(a) Constant $\eta_t = \eta_0$    (b) Schedule $\eta_t = \eta_0 \cdot t^{-1/2}$    (c) Schedule $\eta_t = \eta_0 \cdot t^{-1}$

Figure 2: Mean excess loss of various SGD schedulers on Problem QWC, $\sigma^2 = d$. Horizon-dependent hyperparameters are still needed, with high sensitivity to perturbations of the noise-dependent $\eta_0$.

The use of schedulers does not eliminate the need to tune the learning rate (see Fig. 2b) and selection of learning rate decrease speed is not trivial (compare with Fig. 2c). Typical prescriptions are tuned to the target horizon $T \in \mathbb{N}$, e.g. with the constant but horizon-dependent rate $\eta_t = C \, \sigma^{-1} \, T^{-1/2}$.

## 2 Stab-SGD: Stochastic Gradient Descent with stability-adapted step-sizes

We build our formal statements in the rigorous formalism of stochastic processes, motivated by the crucial part that the step-sizes $\eta_t$ must depend on the local stability ratio of gradient estimates, which itself is a function of the iterates, therefore the step-sizes are random and must be handled carefully.

We take $(\Omega, \mathcal{A}, \mathcal{P})$ to be a probability space, with a filtration $(\mathcal{F}_n)_{n \in \mathbb{N}}$ of $\mathcal{A}$. A sequence of random variables $(X_n)_{n \in \mathbb{N}}$ is said to be "adapted" to $\mathcal{F}$ if $X_i$ is $\mathcal{F}_i$-measurable for all $i \in \mathbb{N}$.

*Intuition.* The standard informal interpretation is that $\mathcal{F}$ models the passage of time, and $X$ is adapted to $\mathcal{F}$ if $X_i$ is "known" at time $i \in \mathbb{N}$. In our case, if the sequence of iterates $(\theta_n)_n$ is adapted to $\mathcal{F}$, then any deterministic function $Y_t = \phi(\theta_t, \ldots, \theta_0)$ of previous iterates is adapted to $\mathcal{F}$ as well.

**Definition 1** (Stochastic Gradient Descent, with unbiased gradients and stochastic stepsizes). A stochastic gradient descent of $\mathcal{L} : \mathbb{R}^d \to \mathbb{R}$ is an $\mathcal{F}$-adapted sequence of random variables $(\theta_n \in \mathbb{R}^d)_{n \in \mathbb{N}}$ together with two $\mathcal{F}$-adapted sequences $(G_n \in \mathbb{R}^d)_{n \in \mathbb{N}}$ and $(\eta_n \in \mathbb{R}_+)_{n \in \mathbb{N}}$, such that for all $t \in \mathbb{N}$, it holds $\theta_{t+1} = \theta_t - \eta_t \cdot G_{t+1}$ and $\mathbb{E}\left[G_{t+1} \mid \mathcal{F}_t\right] = \nabla \mathcal{L}(\theta_t)$

**Definition 2** (Conditional Stability Ratio). The Stability Ratio of a random variable $X \in \mathbb{R}^d$ conditionally on $\mathcal{F}_t$ is defined for any $t \in \mathbb{N}$ as: $\mathrm{Stab}\left(X \mid \mathcal{F}_t\right) = \left\| \mathbb{E}\left[X \mid \mathcal{F}_t\right] \right\|_2^2 \big/ \mathbb{E}\left[ \left\|X\right\|_2^2 \mid \mathcal{F}_t\right]$.

### 2.1 Stab-SGD: A noise-adaptive algorithm with stability oracles

The Stab-SGD iterates $(\theta_t \in \mathbb{R}^d)_{t \in \mathbb{N}}$ of loss $\mathcal{L} : \mathbb{R}^d \to \mathbb{R}$ are defined[3] as

$$\theta_{t+1} = \theta_t - \eta_t \cdot G_{t+1} \qquad \qquad \eta_t = \frac{1}{\beta} \mathrm{Stab}\left(G_{t+1} \mid \mathcal{F}_t\right)$$

for any adapted sequence $(G_t)_t$ satisfying $\mathbb{E}\left[G_{t+1} \mid \mathcal{F}_t\right] = \nabla \mathcal{L}(\theta_t)$ and $\mathbb{V}\left[G_{t+1} \mid \mathcal{F}_t\right] < +\infty$.

Note that Stab-SGD has a single hyperparameter $\beta \in \mathbb{R}_+$, which must be set below the smoothness constant of $\mathcal{L}$. There is no noise-hyperparameter and no horizon-hyperparameter, contrary to SGD bounds typically[4] using step-size $\eta_t \propto \sigma^{-1}/\sqrt{T}$ to give bounds at horizon $T \in \mathbb{N}$ under variance $\sigma^2$. Stab-SGD is a **noise-adaptive** algorithm (conditionally on access to stability ratios), in the sense that it depends on the realized noise level only through the stability ratio, which can be adaptively estimated at every step. This single algorithm adaptively achieves all the convergence rates of Table 1.

Table 1: Convergence rate of Stab-SGD under affine variance $\mathbb{V}\left[G_{t+1} \mid \mathcal{F}_t\right] \le \alpha \|\nabla \mathcal{L}(\theta_t)\|_2^2 + \sigma^2$.

| Noise | $\mathbb{E}\left[\mathcal{L}(\theta_{T+1})\right] - \mathcal{L}^\star$ | | $\mathbb{E}\left[\frac{1}{T}\sum_{t<T}\|\nabla\mathcal{L}(\theta_t)\|_2^2\right]$ |
|---|---|---|---|
| | Convex $\beta$-smooth | $\mu$-strongly convex $\beta$-smooth | Non-convex $\beta$-smooth |
| $\sigma^2 = 0$ | $\mathcal{O}\left(T^{-1}\right)$ | $\mathcal{O}\left(\exp\left(-\frac{1}{1+\alpha}\frac{\mu}{\beta}T\right)\right)$ | $\mathcal{O}\left(T^{-1}\right)$ |
| $\sigma^2 > 0$ | $\mathcal{O}\left(T^{-1/3}\right)$ | $\mathcal{O}\left(T^{-1}\right)$ | $\mathcal{O}\left(T^{-1/2}\right)$ |

Rates in Table 1 are presented *in expectation* for the *last iterate*. In particular, the $\mathcal{O}(T^{-1/3})$ rate in the weakly-convex smooth setting matches Bach and Moulines [2011, Theorem 4] (conjectured to be the optimal horizon-free last-iterate rate for SGD with schedule $\eta_t = \eta_0 \, t^\kappa$ and achieved for $\kappa = -2/3$, see reference for details[5]). The weakly-convex case additionally assumes that there exists $\theta^\star \in \mathbb{R}^d$ such that $\mathcal{L}(\theta^\star) = \mathcal{L}^\star = \inf \mathcal{L}$, see Theorem 1 for the complete statement.

### 2.2 Estimations of Stability Ratio from samples

A natural estimator for $\mathrm{Stab}\left(X\right)$ consists in replacing expectations with averages over $n$ iid samples. Unfortunately, this estimator is strongly biased towards 1 when the number of samples is small. We thus propose another estimator using *Jackknife* resampling for the numerator [Quenouille, 1956].

---

[3]Without loss of generality, we can assume that no $G_{t+1}$ is identically zero by skipping such iterations.

[4]See Garrigos and Gower [2023, Thm 5.5] after canceling gradients with respect to step-size.

[5]A slighly altered $\eta_t = \min(1/2\beta, \eta_0/\sqrt{t})$ was shown to break this conjecture in Liu and Zhou [2023], reaching improved rate $\mathcal{O}(\log(T)/\sqrt{T})$. But it does not reach the $\sigma = 0$ or $\mu > 0$ fast rates without modifying $\eta_t$. Thus the question of getting improved rate for the bottom-left case while retaining adaptivity is left open.

**Definition 3.** The Jackknife estimator of $\mathrm{Stab}\,(X)$ from iid samples $(X_i \in \mathbb{R}^d)_{i \in [n]}$ is

$$R_n = \frac{1}{n-1} \frac{\sum_i \sum_{j \neq i} \langle X_i, X_j \rangle}{\sum_i \|X_i\|_2^2}$$

This can be computed by constructing the sequences $(M_i \in \mathbb{R}^d)_{i \in [n+1]}$ and $(Z_i \in \mathbb{R}^d)_{i \in [n+1]}$ from $M_0 = 0 \in \mathbb{R}^d$ and $Z_0 = 0 \in \mathbb{R}$, as $M_{i+1} = M_i + (X_i - M_i)/(i+1)$ to compute the mean, and $Z_{i+1} = Z_i + (\|X_i\|_2^2 - Z_i)/(i+1)$ for the second moment, then $R_n = \frac{n}{n-1}(\|M_n\|_2^2 - Z_n)/Z_n$. This gives a numerically stable algorithm with $\mathcal{O}(1)$ space complexity to estimate the stability ratio.

**Lemma 1** (Relative error of stability estimation). *Let* $(X_i \in \mathbb{R}^d)_{i \in [n]}$ *be iid random variables. Define* $J_n = \frac{1}{n(n-1)} \sum_{i \neq j} X_i \cdot X_j \in \mathbb{R}$ *and* $Z_n = \frac{1}{n} \sum_i \|X_i\|_2^2 \in \mathbb{R}_+$, *then* $R_n = clip_{[0,1]}(J_n/Z_n) \in [0,1]$.

*Let* $\mu = \mathbb{E}\,[X]$, *and* $\sigma^2 = \mathbb{E}\left[\|X - \mu\|_2^2\right]$, *and* $\kappa = \mathbb{E}\left[\|X\|_2^4\right] / \mathbb{E}\left[\|X\|_2^2\right]^2$. *If* $R_\star = \mathrm{Stab}\,(X) \neq 0$,

$$\mathbb{E}\left[\left(\frac{R_n - R_\star}{R_\star}\right)^2\right] \leq R_\star^{-1} \frac{44 + 4\kappa}{n-1} + R_\star^{-2} \exp\left(-\frac{n}{8\kappa}\right)$$

In particular (when clipping to $[0,1]$), $R_n \to R^\star = \mathrm{Stab}\,(X)$ with high probability, so this estimator is consistent. This lemma is a direct consequence of Lemma A.9. Note that for isotropic multivariate normal random variables $X \in \mathbb{R}^d$, such as $X \sim \mathcal{N}(0, \sigma^2 I)$, it holds[6] $\kappa \leq 1 + 3/d$ (for any $\sigma$). Thus the number of samples needed to estimate a stability ratio $R_\star > 0$ is often of order $n \propto R_\star^{-1}$. The kurtosis[7] $\kappa$ is used to quantify the number of samples needed to estimate the variance.

## 3 Convergence analysis

The tactic used for all following proofs closely tracks the continuous-time analogue by integration along gradient flows (i.e. $[\mathrm{d}\Phi(\mathcal{L}_t) \cdot \partial_t \mathcal{L} \leq -1 \Rightarrow \Phi(\mathcal{L}_t) \leq \Phi(\mathcal{L}_0) - t]$ for any desingularizer $\Phi : \mathbb{R}_+^* \to \mathbb{R}$, such as $\Phi = \log$). This is done by leveraging the "sufficient decrease" inequality[8] $\mathbb{E}\left[\mathcal{L}(\theta_{t+1}) \,|\, \mathcal{F}_t\right] - \mathcal{L}(\theta_t) \leq -\frac{1}{2}\eta_t \|\nabla\mathcal{L}(\theta_t)\|_2^2$ (obtained by construction of Stab-SGD) together with the variance control assumption $\mathbb{V}\left[G_{t+1} \,|\, \mathcal{F}_t\right] \leq \alpha \|\nabla\mathcal{L}(\theta_t)\|_2^2 + \sigma^2$, to obtain an "average sufficient decrease" inequality $\mathbb{E}\left[\mathcal{L}(\theta_{t+1}) - \mathbb{E}\left[\mathcal{L}(\theta_t)\right]\right] \leq -\frac{1}{2\beta}\psi\left(\mathbb{E}\left[\|\nabla\mathcal{L}(\theta_t)\|_2^2\right]\right)$, for a well-chosen convex and increasing function $\psi$, namely $\psi : u \mapsto u^2/(\sigma^2 + (1+\alpha)u)$ for this affine variance control.

This result can be composed with any bound of the form $\mathbb{E}\left[\|\nabla\mathcal{L}(\theta_t)\|_2^2\right] \geq \varphi(\mathbb{E}\left[\mathcal{L}(\theta_t)\right] - \mathcal{L}^\star)$, to bound the optimization gap $\Delta_t = \mathbb{E}\left[\mathcal{L}(\theta_t)\right] - \mathcal{L}^\star$ as $\Delta_t \leq \Phi^{-1}\left(\Phi(\Delta_0) + t/(2\beta)\right)$, where $\Phi$ is obtained by integration of $\mathrm{d}\Phi(u) = 1/(\psi \circ \varphi)(u)$. Different assumptions, leading to various choices of $\varphi$, yield different convergence speeds, as integrated into the function $\Phi$. In particular, local Kurdyka-Łojasiewicz inequalities $\|\nabla\mathcal{L}(\theta)\|_2^2 \geq \varphi(\mathcal{L}(\theta) - \mathcal{L}^\star)$ for convex functions $\varphi$ immediately satisfy the previous condition in expectation (such as $\varphi(x) = 2\mu x$ for $\mu$-strong convexity).

### 3.1 Convergence statements with stability oracles

**Assumption 1** (Stab-SGD with stability oracle and affinely-bounded variance).
This set of assumptions is satisfied if there are constants $\beta \in \mathbb{R}_+^*, \alpha \in \mathbb{R}_+$ and $\sigma \in \mathbb{R}_+$ such that:

- $\mathcal{L} : \mathbb{R}^d \to \mathbb{R}$ is differentiable and uniformly $\beta$-smooth

- $(\theta_t \in \mathbb{R}^d, G_t \in \mathbb{R}^d, \eta_t \in \mathbb{R}_+^*)_{t \in \mathbb{N}}$ is an SGD of $\mathcal{L}$ (Definition 1)

- $\forall t \in \mathbb{N}, \quad \mathbb{V}\left[G_{t+1} \,|\, \mathcal{F}_t\right] \leq \alpha \|\nabla\mathcal{L}(\theta_t)\|_2^2 + \sigma^2$ (affinely bounded variance)

- $\forall t \in \mathbb{N}, \quad \eta_t = \mathrm{Stab}\,(G_{t+1} \,|\, \mathcal{F}_t)\,/\beta$ (strong stability condition)

In such a case, the sequence of random variables $\theta : \mathbb{N} \to \mathbb{R}^d$ are called Stab-SGD iterates.

---

[6]See Lemma A.10 in appendix.

[7]Variables with low kurtosis $\kappa := \mathbb{E}\left[\|X\|_2^4\right] / \mathbb{E}\left[\|X\|_2^2\right]^2 = 1/\mathrm{Stab}\left(\|X\|_2^2\right)$ have empirical estimates of variance close to true variance, while high kurtosis requires more samples for accurate estimation of variance.

[8]See Beck [2014, Lemma 4.3 and Sec 4.7.3] for the classical deterministic analysis leveraging this condition.

The following theorems are derived from Corollary A.1, Corollary A.2, and Proposition A.3.

**Theorem 1** (Weakly convex smooth rate). *If $\mathcal{L} : \mathbb{R}^d \to \mathbb{R}$ is convex, uniformly $\beta$-smooth, and if there exists $\theta^\star \in \Theta$ such that $\mathcal{L}^\star = \mathcal{L}(\theta^\star)$, then for any Stab-SGD iterates $\theta : \mathbb{N} \to \mathbb{R}^d$ satisfying Assumption 1, and if*

$$T \geq \frac{2}{3} \frac{\beta D_0^4 \sigma^2}{\varepsilon^3} + (1 + \alpha) \frac{\beta D_0^2}{\varepsilon}$$

*then $\mathbb{E}\left[\,\mathcal{L}(\theta_{T+1})\,\right] \leq \mathcal{L}^\star + \varepsilon$, where $D_0^2 = \mathbb{E}\left[\,\|\theta_0 - \theta^\star\|_2^2\,\right]$ measures initial distance to optimum.*

**Theorem 2** (Strongly convex smooth rate). *If $\mathcal{L} : \mathbb{R}^d \to \mathbb{R}$ is uniformly $\beta$-smooth and $\mu$-strongly convex, then any Stab-SGD iterates $\theta : \mathbb{N} \to \mathbb{R}^d$ satisfying Assumption 1, and if*

$$T \geq \frac{\sigma^2 \beta}{2\mu^2\varepsilon} + (1 + \alpha)\frac{\beta}{\mu}\log\left(\frac{\Delta_0}{\varepsilon}\right)$$

*then $\mathbb{E}\left[\,\mathcal{L}(\theta_{T+1})\,\right] \leq \mathcal{L}^\star + \varepsilon$, where $\Delta_0 = \mathbb{E}\left[\,\mathcal{L}(\theta_0)\,\right] - \mathcal{L}^\star$ measures the initial optimization gap.*

**Theorem 3** (Non-convex rate). *If $\mathcal{L} : \mathbb{R}^d \to \mathbb{R}$ is uniformly $\beta$-smooth, for any Stab-SGD iterates $\theta : \mathbb{N} \to \mathbb{R}^d$ satisfying Assumption 1,*

$$\forall T \in \mathbb{N}, \quad \mathbb{E}\left[\frac{1}{T}\sum_{t<T}\|\nabla\mathcal{L}(\theta_t)\|_2^2\right] \leq (1+\alpha)\frac{2\beta\Delta_0}{T} + \sqrt{\frac{2\beta\Delta_0\sigma^2}{T}}$$

*where $\Delta_0 = \mathbb{E}\left[\,\mathcal{L}(\theta_0)\,\right] - \mathcal{L}^\star$ measures the optimization gap at initialization.*

## 3.2 Inline Stability Estimation

To incorporate the estimation of the gradient's stability ratio in the algorithm at little overhead cost, we propose Algorithm 1, with access to noisy gradients but without a stability oracle.

This algorithm uses three parameters to control estimation overhead:

- a sample overhead $\zeta \in \mathbb{R}_+^*$ (of order 10 to 100)
- a time step $\kappa \in \mathbb{R}_+^*$
- a time exponent $\gamma \in [0, 1]$

The most conservative configuration ($\gamma = 0$, $\kappa = 1$) estimates the stability ratio at every step. However, if the stability ratio is expected to be relatively continuous, then a looser configuration ($\gamma = 1$) will perform only logarithmically many estimations with respect to the horizon, which is a minimal overhead.

In looser configurations, incorrect ratios could yield temporary saturations (overestimation), or temporary slowdowns (underestimation).

**Input :** $x_0 \in \mathbb{R}^d, \eta_0 \in \mathbb{R}_+^*, \zeta \geq 1, \kappa \in \mathbb{R}_+^*, \gamma \in [0, 1]$
$(S, T) \leftarrow (S_0 = 1 \in [0, 1], T_0 = 1 \in \mathbb{R}_+^*)$
**for** $k \in \mathbb{N}$ **do**
  **if** $(k = 0)$ *or* $(k \geq T)$ **then**
    $n \leftarrow \lceil \zeta/S \rceil \in \mathbb{N}$
    $(M_0, Z_0) \leftarrow (0 \in \mathbb{R}^d, 0 \in \mathbb{R})$
    **for** $i \in [n]$ **do**
      $v_i \leftarrow G_{k,i} \in \mathbb{R}^d$  [estimate of $\nabla\mathcal{L}(x_k)$]
      $M_{i+1} \leftarrow M_i + (v_i - M_i)/(i+1)$
      $Z_{i+1} \leftarrow Z_i + (\|v_i\|_2^2 - Z_i)/(i+1)$
    **end**
    $S \leftarrow \dfrac{n}{n-1}\dfrac{\|M_i\|_2^2 - Z_i}{Z_i}$   [Stab estimator]
    $T \leftarrow T + \kappa \cdot T^\gamma$
  **end**
  $m_k \leftarrow G_k \in \mathbb{R}^d$  [fresh estimate of $\nabla\mathcal{L}(x_k)$]
  $x_{k+1} \leftarrow x_k - (\eta_0 \cdot S) \cdot m_k$
**end**

**Algorithm 1:** Inline Stab-SGD

While we can't guarantee the quality of the looser configurations without additional assumptions such as continuity of noise variance, we observe empirically that loose options such as ($\gamma = 1$, $\kappa = 0.5$) still display all key properties of Stab-SGD: no intial plateau or explosion, and no saturation.

*Note on overhead cost.* For a total of $T$ gradients queried at stability ratios above $s_\star > 0$, at most a fraction $c/(1+c) \in \,]0, 1[$ of queries are dedicated to stability estimation, where $c \in \mathbb{R}_+$ can be controlled by tuning $\kappa$ (e.g. set to $c \leq 1$). If $\gamma = 1$, then $c \leq \zeta\, s_\star^{-1}\kappa^{-1}\log(T)/T$ is vanishing with $T$. If $\gamma = 0$, then $c \leq \zeta\, s_\star^{-1}\kappa^{-1}$. For the exponent $\alpha$, the movement's characteristic timescale is estimated using $\mathbb{E}\left[\|G_{t+k+1}\|\,|\,\mathcal{F}_t\right] \leq (\|\nabla\mathcal{L}(\theta_t)\|^2 + \sigma^2)^{1/2}$ (unrigorously) without expectations for a quick approximation, and smoothness as $\|\nabla\mathcal{L}(x)\|_2^2 \leq 2\beta(\mathcal{L}(x) - \mathcal{L}^\star)$ with $\mathcal{L}^\star = 0$ for simplicity,

$$\|\theta_{t+\Delta t} - \theta_t\| \leq \sum_{k\leq\Delta t}\eta_t\|G_{t+k+1}\| \lessapprox \sum_{k\leq\Delta t}\frac{1}{\beta}\frac{\|\nabla\mathcal{L}(\theta_{t+k})\|^2}{(\|\nabla\mathcal{L}(\theta_{t+k})\|^2 + \sigma^2)^{1/2}} \leq \frac{2}{\sigma}\sum_{k\leq\Delta t}\mathcal{L}(\theta_{t+k})$$

If $\mathcal{L}(\theta_t) \leq C_0 t^{-1/3}$, this bound is at most $C_1 \Delta t \cdot t^{-1/3}$, so the unit-scale movements' characteristic time is at most $\Delta t \approx C_1^{-1} t^{1/3}$. This quick calculation suggests that even in the worst case, $\gamma = 1/3$ should remain a safe option. Similarly, a rate $\mathcal{L}(\theta_t) \leq C_0 t^{-1}$ could use $\gamma = 1$ safely, but we conjecture that such loose settings will be useable far outside this regime. Characterisation of precise noise-continuity hypotheses under which such choices are provably safe is left for future work.

## 4 Experiments

**Methods.**[9] We perform experiments in two stages, first training for $T_0 \in \mathbb{N}$ (tuning horizon) iterations on a grid of hyperparameters ($\log \eta_0$ from -7 to +5 by increments of 0.5, a total of $k = 25$ values). We then select the best hyperparameter (at $T_0$) and train with this value for $T \in \mathbb{N}$ iterations. The fraction of the total budget spent on hyperparameter tuning is thus $k\, T_0/(k\, T_0 + T)$, and the tuning overhead (excess cost of tuning relative to training) is $k\, T_0/T$. These quantities are rarely reported on large-scale experiments failing to take hyperparameter-tuning costs into account, but there is a common intuition that popular algorithms require a massive fraction of budget allocated to tuning.

### 4.1 Comparisons with concurrent schedule-free algorithms

**Cheap regime: low noise, strong convexity.** Fig. 3 presents loss as a function of tuning horizon.

Vertical gaps within curves indicate the final gap in loss if less budget is spent on tuning.

The sensitivity of SGD is visible on the right (the noise-dominated regime). The long-horizon optimal learning rate cannot be selected well on short tuning horizons (which do not enter the noise regime), a property that is likely shared by deep learning settings.

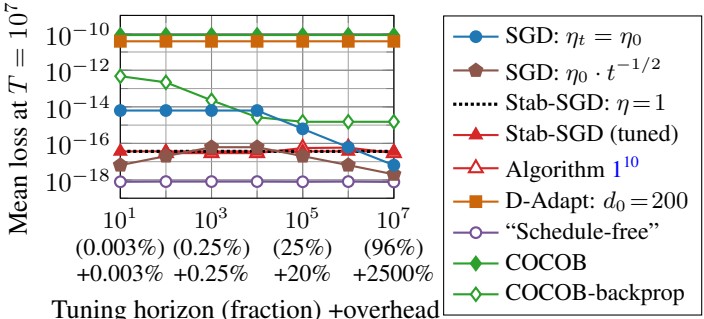

Figure 3: Mis-tuning cost on Problem QSC, $\sigma^2 = 10^{-16} \cdot d$.

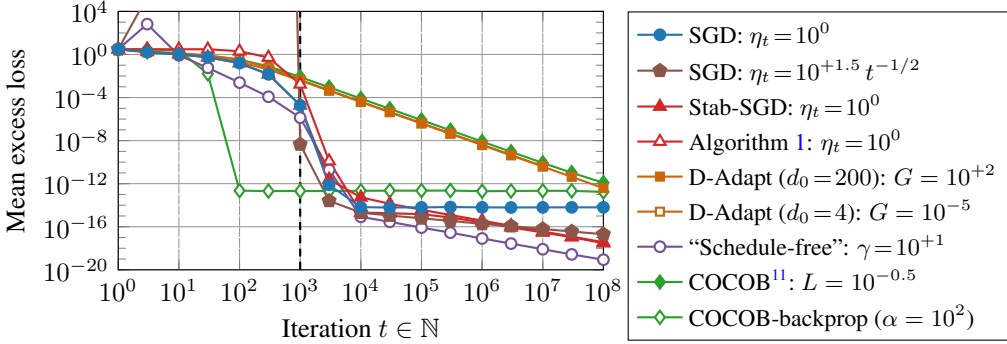

Figure 4: Evolution of the loss on Problem QSC, $\sigma^2 = 10^{-16} \cdot d$. Tuning horizon as dashed line.

Figure 4 depicts the evolution of the loss over time for a tuning horizon at $T_0 = 10^3$. Algorithms designed for the noisy regime alone (such as COCOB and D-Adapt, which use iterate-averaging) fail to take advantage of strong convexity, leaving them 8 orders of magnitude behind at $10^4$ iterations.

---

[9]The source code to reproduce all experiments of this section and the next is available online at https://www.github.com/robindar/2025-NeurIPS_Stab-SGD.

[10]Algorithm 1 with (loose) $\gamma = 1$, $\kappa = 1$, and $\zeta = 50$. Iteration count is total number of gradients queried. Results overlap with Stab-SGD (with stability oracles), both tuned and pre-set to $\eta = \beta^{-1}$, hardly visible.

[11]Results overlap with D-Adapt (both settings). Both COCOB and D-Adapt are average-iterate algorithms, the averaging slows down convergence in this regime, yielding very similar speeds.

**Expensive regime: smooth with high noise.** Figure 5 presents mean loss as a function of tuning horizon for Problem QWC. Each training run at $10^7$ iterations takes about one hour on our CPUs.

Slope indicates sensitivity of the hyperparameter to the tuning horizon. Algorithms with large slopes are only usable if essentially all budget is spent tuning the sensitive parameter.

At high noise with this training horizon ($10^7$), SGD only outperforms Stab-SGD if at least 71% of the total budget is spent on hyperparameter tuning, i.e. if an extra +250% of the training budget is spent tuning at $T_0 = 10^6$ horizon.

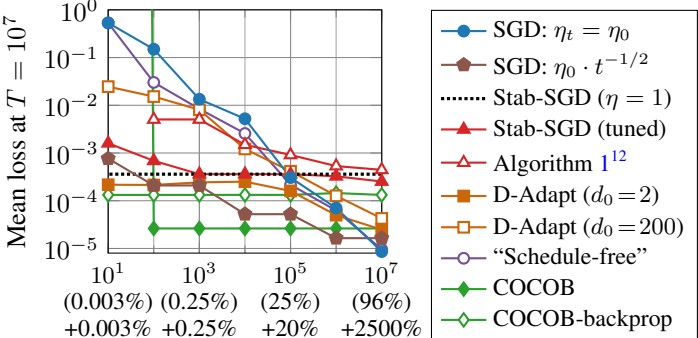

Figure 5: Mis-tuning cost on Problem QWC, $\sigma^2 = d$.

Algorithms previously well-performing (such as "Schedule-Free SGD") are not as good in this regime, sometimes even indistinguishable from equivalently-tuned SGD. On the contrary, algorithms designed for this setting (e.g. COCOB) perform much better. This leaderboard reversal induces a difficulty to choose the best algorithm with unknown noise level. Stab-SGD gives consistent performance in both settings. The price of this adaptivity is apparent in both cases, but not necessarily prohibitive.

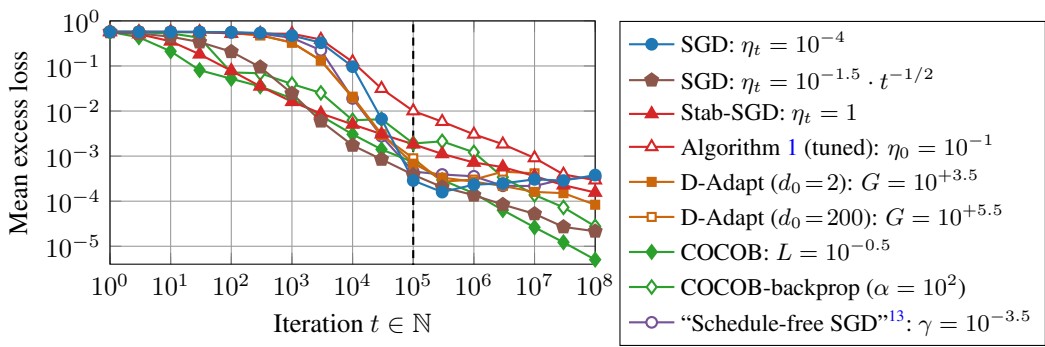

Figure 6: Evolution of the loss on Problem QWC, $\sigma^2 = d$. Tuning horizon as dashed line.

Although its asymptotic performance is slightly suboptimal compared to other methods, and does not achieve the minimax optimal rate of averaging methods, the complete absence of noise-dependent tuning of the hyperparameter, and reasonable properties (no plateau, no explosion, no saturation) of the Stab-SGD trajectory make it an interesting research direction for schedule-free settings aiming for those properties, particularly when the hyperparameter-tuning cost is taken into account.

The proof of last-iterate expected loss matching these observations also highlights the importance of the stability ratio of gradients in the development of smooth optimization with last-iterate guarantees, possibly better suited to the study of non-convex models such as neural networks.

We conjecture that it will be possible to construct accelerated noise-adaptive algorithms which will be competitive not only on low tuning budgets, but also on high tuning budgets (right end of Figure 5) where Stab-SGD and its stability-oracle-free variant Algorithm 1 are found to be lacking, possibly due to a suboptimal asymptotic rate. Nonetheless, works on accelerated stochastic algorithms typically use hyperparameters with convoluted dependence on noise parameters, see for instance Jain et al. [2018, Thm 1] with impressive speed but *four* noise-dependent hyperparameters for the case of quadratic problems alone. Therefore, we suspect that an accelerated noise-adaptive horizon-free extension of Stab-SGD could be a vastly more complicated algorithm than the ones presented here.

---

[12]Algorithm 1 with (loose) $\gamma = 1$, $\kappa = 1$, and $\zeta = 50$. Iteration count is total number of gradients queried.

[13]Results almost perfectly overlap with SGD, difference hardly visible

## 4.2 ResNet training experiments on CIFAR-10

**Methods.** We perform experiments on the CIFAR-10 image classification dataset [Krizhevsky, 2009] with the ResNet-56 architecture[14] [He et al., 2015a, Sec 4.2]. We compare with the aforementioned original ResNet publication, which uses a learning rate $10^{-1}$ for 32k iterations, then $10^{-2}$ for the next 16k and $10^{-3}$ for the last 16k, totaling 64k iterations (thresholds depicted by dashed vertical lines). We use batches of size 128 sampled without replacement for each epoch (391 batches / epoch). We restrict the hyperparameter search for $\log_{10}(\eta_0)$ to a grid from $-3$ to $+1$ by steps of 0.5, informed by choices in the original reference. We use an $\ell_2^2$ weight decay with $\lambda = 10^{-4}$ for all runs.

We run Algorithm 1 with $\eta_0 = 10^{+1}$, with the configuration $\kappa = 10^{-1}$, $\gamma = 1$ and $\zeta = 100$. To evaluate the overhead cost of stability-estimation, we provide both curves: *oracle* where the number of iterations is the number of weight-updating steps (*effective* iterations); and *raw* where iterations corresponds to the total number of gradients queried, including gradients used for stability estimation.

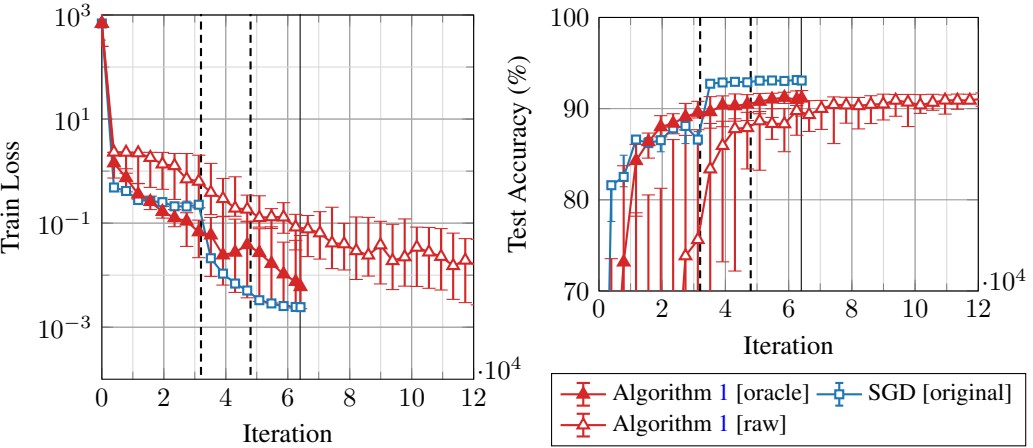

Figure 7: ResNet-56 on CIFAR-10. Evolution of accuracy and loss, presented as medians and quartiles for error bars, for 20 seeds of Algorithm 1. Average runtime of 4h to 5h per seed on GPU.

The results presented in Figure 7 show performance comparable between the *oracle* variant and SGD with tuned schedule. Without the need to tune a scheduler, this algorithm has correctly used a first large step-size then much lower, allowing it to break past the mid-training plateau incurred by SGD (visible at 32k iterations). Nonetheless, the variance across seeds is significantly increased, and taking into account the cost of stability-ratio estimation (with the *raw* variant) we can estimate that it needs on the order of twice as many iterations for similar performance in this experiment. For context, the choice of scheduler must have been guided by experiments, say $k \in \mathbb{N}^*$ runs[15], thus the total cost comparison with noise-dependent scheduler tuning is between $k \times T$ for the scheduled SGD, and $2\,T$ for Algorithm 1 (*raw*), which is in favor of the adaptive algorithm presented here as soon as $k > 2$.

Although not competitive on such problems at this stage of development, Alg. 1 remains a promising research direction, since it maintains in this non-convex setting the desired properties: no initial explosion or plateau, and no saturation requiring large learning rate shrinkage. It reaches lower loss than SGD with $\eta = 10^{-1}$ (before first threshold) without tuning a threshold (at 32k) or shrinking factor ($\times 0.1$).

Fig. 8 shows evolution of the Stability Ratio along trajectories. Shrinkage behavior is consistent with the original: small variations up to $10^4$ then decreasing by several orders of magnitude. The original tuned schedule shrinked learning rates at 32k and 48k. More details in Appendix C.2.

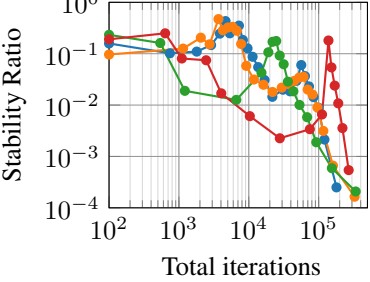

Figure 8: Stability along trajectory

---

[14]Note that the numbering refers to the CIFAR-targeting architectures [He et al., 2015a, Sec 4.2], contrary to the much larger ResNet-18 and ResNet-30 [He et al., 2015a, Sec 4.1], which target ImageNet [Deng et al., 2009].

[15]The number of tuning runs $k \in \mathbb{N}^*$ is not given in the original reference, and left for the reader to estimate.

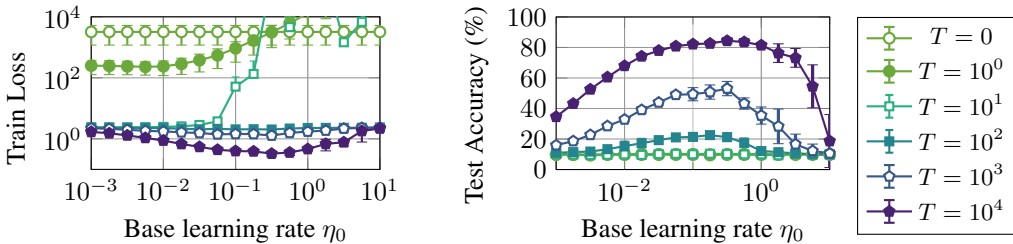

Figure 9: ResNet-56, loss and accuracy as a function of learning rate for SGD.

This is consistent with convex experiments, indicating that Stab-SGD enables selection of a larger base learning rate, which is automatically adapted to the noise level. Indeed, with the initial Stability Ratio around $10^{-1}$, the effective learning rate of the first $10^3$ iterations is around $10^0$, which is not far from the optimum observed for SGD over that period (see Fig. 9). The performance of the *oracle* variant (i.e. ignoring stability-estimation costs) showcases the competitive behavior that could be reachable for future works achieving cheaper stability estimations.

**Conclusion.** We introduced the Stability Ratio, a natural measure of local relative noise of stochastic gradient estimates, yielding a schedule-free variant of SGD achieving real adaptivity to the noise level. We presented new theoretical tools to analyze this stochastic-step algorithm in convex, strongly convex and non-convex settings, with strong last-iterate guarantees in expectation, obtained by a stochastic version of Kurdyka-Łojasiewicz integration. We validated the adaptivity of this proposed algorithm with convex experiments showing that it outperforms algorithms not achieving the fast rate on strongly convex problems (such as COCOB or D-Adapt, developed for less regular settings), and that it remains in the competitive range without the need for a noise-dependent tuning of hyperparameters. We measured performance on ResNet networks for CIFAR-10 which further strenghtened that when taking hyperparameter-tuning budgets into account, this last-iterate noise-adaptive algorithm retains reasonable performance on non-convex deep learning problems. This shows that future algorithms leveraging this idea together with improved estimates of the stability ratio along a training trajectory will likely be able to outperform extensively-tuned learning rate schedulers in deep learning scenarios.

## Acknowledgements

This work was supported by the French government managed by the Agence Nationale de la Recherche (ANR) through France 2030 program with the reference ANR-23-PEIA-005 (REDEEM project). It was also funded in part by the Groupe La Poste, sponsor of the Inria Foundation, in the framework of the FedMalin Inria Challenge.

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

# A  Appendix

In all the appendix, $\mathcal{L} : \mathbb{R}^d \to \mathbb{R}^d$ is a $\beta$-smooth function for some $\beta \in \mathbb{R}_+^*$, and $(\theta_t \in \mathbb{R}^d)_{t \in \mathbb{N}}$ is a stochastic gradient descent of $\mathcal{L}$ (Definition 1) with gradient estimates $(G_{t+1} \in \mathbb{R}^d)_{t \in \mathbb{N}}$ and step-sizes $(\eta_t \in \mathbb{R}_+^*)_{t \in \mathbb{N}}$ satisfying $\theta_{t+1} = \theta_t - \eta_t \cdot G_{t+1}$ and $\mathbb{E}\left[ G_{t+1} \,|\, \mathcal{F}_t \right] = \nabla \mathcal{L}(\theta_t)$, where $\mathcal{F}$ is the time filtration.

When appropriate, the variable $T \in \mathbb{N}$ denotes a horizon, $\mathcal{L}^\star = \inf \mathcal{L}$ is the infimum of the loss, and $\theta^\star \in \mathbb{R}^d$ is a global optimum $\mathcal{L}(\theta^\star) = \mathcal{L}^\star$ when it is assumed to exist.

## A.1  Rates with stability oracle

**Lemma A.1** (Base reduction).
*If for all $t \leq T$, it holds $\eta_t \leq \beta^{-1}\,\mathrm{Stab}\left( G_{t+1} \,|\, \mathcal{F}_t \right)$ (weak stability condition), then it holds*

$$\forall t \leq T, \quad \mathbb{E}\left[ \mathcal{L}(\theta_{t+1}) \,|\, \mathcal{F}_t \right] \leq \mathcal{L}(\theta_t) - \frac{\eta_t}{2} \|\nabla \mathcal{L}(\theta_t)\|_2^2$$

*Proof.* By $\beta$-smoothness of $\mathcal{L}$, then simplifying conditional expectations,

$$\mathcal{L}(\theta_{t+1}) \leq \mathcal{L}(\theta_t) - \eta_t \, \nabla \mathcal{L}(\theta_t) \cdot G_{t+1} + \frac{1}{2} \beta \eta_t^2 \|G_{t+1}\|_2^2$$

$$\mathbb{E}\left[ \mathcal{L}(\theta_{t+1}) \,|\, \mathcal{F}_t \right] \leq \mathbb{E}\left[ \mathcal{L}(\theta_t) - \eta_t \, \nabla \mathcal{L}(\theta_t) \cdot G_{t+1} + \frac{1}{2} \beta \eta_t^2 \|G_{t+1}\|_2^2 \,\Big|\, \mathcal{F}_2 \right]$$

$$\leq \mathcal{L}(\theta_t) - \eta_t \, \nabla \mathcal{L}(\theta_t) \cdot \mathbb{E}\left[ G_{t+1} \,|\, \mathcal{F}_t \right] + \frac{1}{2} \beta \eta_t^2 \, \mathbb{E}\left[ \|G_{t+1}\|_2^2 \,|\, \mathcal{F}_t \right]$$

$$\leq \mathcal{L}(\theta_t) - \eta_t \, \|\nabla \mathcal{L}(\theta_t)\|_2^2 + \frac{1}{2} \beta \eta_t^2 \, \frac{\|\nabla \mathcal{L}(\theta_t)\|_2^2}{\mathrm{Stab}\left( G_{t+1} \,|\, \mathcal{F}_t \right)}$$

$$\leq \mathcal{L}(\theta_t) - \eta_t \, \|\nabla \mathcal{L}(\theta_t)\|_2^2 + \frac{1}{2} \eta_t \, \|\nabla \mathcal{L}(\theta_t)\|_2^2$$

$$\leq \mathcal{L}(\theta_t) - \frac{\eta_t}{2} \, \|\nabla \mathcal{L}(\theta_t)\|_2^2$$

$\square$

**Lemma A.2.** *For all $(\sigma, \alpha) \in \mathbb{R}_+^2$, the function $\psi : \mathbb{R}_+ \to \mathbb{R}_+$ is strictly increasing and convex.*

$$\psi : u \mapsto \frac{u^2}{\sigma^2 + (1 + \alpha)u}$$

*Proof.* By continuity at zero and twice-differentiability of $\psi$ on $\mathbb{R}_+^*$, it suffices to check, for every $u \in \mathbb{R}_+^*$, that $\mathrm{d}\psi(u) > 0$ (strict increase) and $\mathrm{d}^2\psi(u) \geq 0$ (convexity). Write $c = 1 + \alpha$ and compute

$$\mathrm{d}\psi(u) = \frac{2u(\sigma^2 + cu) - cu^2}{(\sigma^2 + cu)^2} = \frac{2u\sigma^2 + cu^2}{(\sigma^2 + cu)^2} > 0$$

Then the second derivative of $\psi$ is observed to be non-negative, which concludes the proof.

$$\mathrm{d}^2\psi(u) = \frac{(2\sigma^2 + 2cu)(\sigma^2 + cu)^2 - (2u\sigma^2 + cu^2) \cdot 2c(\sigma^2 + cu)}{(\sigma^2 + cu)^4}$$

$$= \frac{(2\sigma^2 + 2cu)(\sigma^2 + cu) - 2c(2u\sigma^2 + cu^2)}{(\sigma^2 + cu)^3}$$

$$= \frac{2\sigma^4 + 4\sigma^2 cu + 2c^2u^2 - (4\sigma^2 cu + 2c^2u^2)}{(\sigma^2 + cu)^3} = \frac{2\sigma^4}{(\sigma^2 + cu)^3} \geq 0$$

$\square$

**Lemma A.3.** *For all $(\sigma, \alpha) \in \mathbb{R}_+^2$, the function $\psi : u \in \mathbb{R}_+ \mapsto u^2 \cdot (\sigma^2 + (1 + \alpha)\,u)^{-1}$ admits an inverse $\psi^{-1} : \mathbb{R}_+ \to \mathbb{R}_+$, and for all $x \in \mathbb{R}_+$, it holds $\psi^{-1}(x) \leq (1 + \alpha)\,x + \sigma\,\sqrt{x}$.*

*Proof.* By Lemma A.2, $\psi$ is strictly increasing and has $\psi(u) \to 0$ when $u \to 0$, and $\psi(u) \to +\infty$ when $u \to \infty$, therefore $\psi$ is bijective and admits an inverse, which is also strictly increasing.

Moreover, defining $z = (1+\alpha)x + \sigma\sqrt{x}$, observe that

$$\psi(z) = \frac{z^2}{\sigma^2 + (1+\alpha)z} = \frac{(1+\alpha)x^2 + 2(1+\alpha)x\sqrt{x} + \sigma^2 x}{\sigma^2 + (1+\alpha)^2 x + (1+\alpha)\sigma\sqrt{x}}$$

$$= x + \frac{(1+\alpha)x\sqrt{x}}{\sigma^2 + (1+\alpha)^2 x + (1+\alpha)\sigma\sqrt{x}} \geq x$$

Therefore $x \leq \psi(z)$, which implies $\psi^{-1}(x) \leq z$ and concludes the proof. $\qquad\square$

**Lemma A.4** (Key reduction)**.**
*If for all $t \leq T$, it holds $\mathbb{V}[G_{t+1} \,|\, \mathcal{F}_t] \leq \alpha\|\nabla\mathcal{L}(\theta_t)\|_2^2 + \sigma^2$ (affinely bounded variance), and $\eta_t = \beta^{-1}\,\mathrm{Stab}\,(G_{t+1} \mid \mathcal{F}_t)$ (strong stability condition), then it holds*

$$\forall t \leq T, \quad \mathbb{E}[\mathcal{L}(\theta_{t+1})] - \mathbb{E}[\mathcal{L}(\theta_t)] \leq -\frac{1}{2\beta}\psi\left(\mathbb{E}\left[\|\nabla\mathcal{L}(\theta_t)\|_2^2\right]\right)$$

*where $\psi : u \mapsto u^2/(\sigma^2 + (1+\alpha)\,u)$ is a convex and increasing function.*

*Proof.* Starting from Lemma A.1, and using the affinely bounded variance asssumption to obtain the inequality $\mathbb{E}\left[\|G_{t+1}\|_2^2 \,\big|\, \mathcal{F}_t\right] \leq \|\mathbb{E}[G_{t+1}\,|\,\mathcal{F}_t]\|_2^2 + \mathbb{V}[G_{t+1}\,|\,\mathcal{F}_t] \leq (1+\alpha)\|\nabla\mathcal{L}(\theta_t)\|_2^2 + \sigma^2$, substituted in the stability ratio, we obtain

$$\mathbb{E}[\mathcal{L}(\theta_{t+1})\,|\,\mathcal{F}_t] - \mathcal{L}(\theta_t) \leq -\frac{\eta_t}{2}\|\nabla\mathcal{L}(\theta_t)\|_2^2$$

$$\leq -\frac{1}{2\beta}\frac{\|\nabla\mathcal{L}(\theta_t)\|_2^2}{\sigma^2 + (1+\alpha)\|\nabla\mathcal{L}(\theta_t)\|_2^2}\|\nabla\mathcal{L}(\theta_t)\|_2^2$$

$$\leq -\frac{1}{2\beta}\psi\left(\|\nabla\mathcal{L}(\theta_t)\|_2^2\right)$$

Therefore, taking expectations, and using convexity of $\psi$ (Lemma A.2) as $\mathbb{E}[\psi(U)] \geq \psi(\mathbb{E}[U])$,

$$\mathbb{E}[\mathcal{L}(\theta_{t+1})] - \mathbb{E}[\mathcal{L}(\theta_t)] \leq -\frac{1}{2\beta}\mathbb{E}\left[\psi\left(\|\nabla\mathcal{L}(\theta_t)\|_2^2\right)\right] \leq -\frac{1}{2\beta}\psi\left(\mathbb{E}\left[\|\nabla\mathcal{L}(\theta_t)\|_2^2\right]\right)$$

$$\square$$

**Lemma A.5** (KŁ stochastic integration)**.**
*If for all $t \leq T$, it holds $\mathbb{V}[G_{t+1}\,|\,\mathcal{F}_t] \leq \alpha\|\nabla\mathcal{L}(\theta_t)\|_2^2 + \sigma^2$ (affinely bounded variance), and $\eta_t = \beta^{-1}\,\mathrm{Stab}\,(G_{t+1} \mid \mathcal{F}_t)$ (strong stability condition), and if it holds for some increasing function $\varphi : \mathbb{R}_+ \to \mathbb{R}_+$ that $\mathbb{E}\left[\|\nabla\mathcal{L}(\theta_t)\|_2^2\right] \geq \varphi\left(\mathbb{E}[\mathcal{L}(\theta_t)] - \mathcal{L}^\star\right)$, then it holds*

$$\forall t \leq T, \quad \mathbb{E}[\mathcal{L}(\theta_{t+1})] - \mathcal{L}^\star \leq \Phi^{-1}\left(\Phi(\mathbb{E}[\mathcal{L}(\theta_0)] - \mathcal{L}^\star) + \frac{t}{2\beta}\right)$$

*where $\Phi : \mathbb{R}_+^* \to \mathbb{R}$ is the[16] function defined as $\mathrm{d}\Phi(u) = -\left(\sigma^2 + (1+\alpha)\,\varphi(u)\right) \cdot \varphi(u)^{-2}$.*

*In particular, if $T \geq 2\beta\left(\Phi(\varepsilon) - \Phi(\Delta_0)\right)$ for $\Delta_0 = \mathbb{E}[\mathcal{L}(\theta_0)] - \mathcal{L}^\star$, then $\mathbb{E}[\mathcal{L}(\theta_{T+1})] \leq \mathcal{L}^\star + \varepsilon$.*

*Proof.* Starting from Lemma A.4, and using the last assumption since $\psi$ is increasing,

$$\mathbb{E}[\mathcal{L}(\theta_{t+1})] - \mathbb{E}[\mathcal{L}(\theta_t)] \leq -\frac{1}{2\beta}\psi\left(\mathbb{E}\left[\|\nabla\mathcal{L}(\theta_t)\|_2^2\right]\right) \leq -\frac{1}{2\beta}(\psi \circ \varphi)\left(\mathbb{E}[\mathcal{L}(\theta_t)] - \mathcal{L}^\star\right)$$

Note that by definition $\mathrm{d}\Phi(u) = -1/(\psi \circ \varphi)(u)$. We will use this to simplify the above equation, but also to observe that $\mathrm{d}\Phi$ is increasing since $(\psi \circ \varphi)$ is increasing as a composition of increasing

---

[16]uniquely defined only up to a constant, the bound is invariant by change of such additive constant

functions. Therefore, $\Phi$ is a convex function (since it has increasing derivative) which can be used as $\Phi(y) - \Phi(x) \geq d\Phi(x) \cdot (y - x)$ to further simplify

$$d\Phi\left(\mathbb{E}\left[\mathcal{L}(\theta_t)\right] - \mathcal{L}^\star\right) \cdot \left(\mathbb{E}\left[\mathcal{L}(\theta_{t+1})\right] - \mathbb{E}\left[\mathcal{L}(\theta_t)\right]\right) \geq \frac{1}{2\beta}$$

$$\Phi\left(\mathbb{E}\left[\mathcal{L}(\theta_{t+1})\right] - \mathcal{L}^\star\right) - \Phi\left(\mathbb{E}\left[\mathcal{L}(\theta_t)\right] - \mathcal{L}^\star\right) \geq \frac{1}{2\beta}$$

Observing that $\Phi$ is decreasing (since it has negative derivate), this implies

$$\mathbb{E}\left[\mathcal{L}(\theta_{t+1})\right] - \mathcal{L}^\star \leq \Phi^{-1}\left(\Phi\left(\mathbb{E}\left[\mathcal{L}(\theta_0)\right] - \mathcal{L}^\star\right) + \frac{t}{2\beta}\right)$$

Defining $\Delta_0 = \mathbb{E}\left[\mathcal{L}(\theta_0)\right] - \mathcal{L}^\star$ and injecting $T \geq 2\beta(\Phi(\varepsilon) - \Phi(\Delta_0))$ in the previous equation yields the final claim, by decrease of $\Phi$.

$$\mathbb{E}\left[\mathcal{L}(\theta_{T+1})\right] - \mathcal{L}^\star \leq \Phi^{-1}\left(\Phi(\Delta_0) + \frac{T}{2\beta}\right) \leq \varepsilon$$

$\square$

**Lemma A.6** (Squared distance to optimum is a submartingale).
*If $\mathcal{L} : \mathbb{R}^d \to \mathbb{R}$ is convex, and there exists $\theta^\star \in \mathbb{R}^d$ such that $\mathcal{L}(\theta^\star) = \mathcal{L}^\star$, and if for all $t \leq T$, it holds $\eta_t \leq \beta^{-1} \operatorname{Stab}(G_{t+1} \mid \mathcal{F}_t)$ (weak stability condition), then it holds*

$$\forall t \leq T, \quad \mathbb{E}\left[\|\theta_t - \theta^\star\|_2^2\right] \leq \mathbb{E}\left[\|\theta_0 - \theta^\star\|_2^2\right]$$

*Proof.* Define the random variable $D_t \in \mathbb{R}$ as $D_t^2 = \|\theta_t - \theta^\star\|_2^2$. Observe that expanding the square,

$$D_{t+1}^2 - D_t^2 = -2\eta_t\, G_{t+1} \cdot (\theta_t - \theta^\star) + \eta_t^2\, \|G_{t+1}\|_2^2$$

Thus taking conditional expectations and using the weak stability condition,

$$\mathbb{E}\left[D_{t+1}^2 \mid \mathcal{F}_t\right] - D_t^2 = -2\eta_t \nabla\mathcal{L}(\theta_t) \cdot (\theta_t - \theta^\star) + \eta_t^2 \|\nabla\mathcal{L}(\theta_t)\|_2^2 / \operatorname{Stab}(G_{t+1} \mid \mathcal{F}_t)$$

$$\leq -2\eta_t \nabla\mathcal{L}(\theta_t) \cdot (\theta_t - \theta^\star) + \frac{\eta_t}{\beta} \|\nabla\mathcal{L}(\theta_t)\|_2^2$$

By convexity of $\mathcal{L}$, the first term can be bounded with $\mathcal{L}^\star - \mathcal{L}(\theta_t) \geq -\nabla\mathcal{L}(\theta_t) \cdot (\theta_t - \theta^\star)$, and the second term can be bounded by $\beta$-smoothness of $\mathcal{L}$ as $\|\nabla\mathcal{L}(\theta_t)\|_2^2 \leq 2\beta(\mathcal{L}(\theta_t) - \mathcal{L}^\star)$, thus

$$\mathbb{E}\left[D_{t+1}^2 \mid \mathcal{F}_t\right] - D_t^2 \leq -2\eta_t\left(\mathcal{L}^\star - \mathcal{L}(\theta_t)\right) + 2\eta_t\left(\mathcal{L}(\theta_t) - \mathcal{L}^\star\right) \leq 0$$

Hence $\mathbb{E}\left[D_{t+1}^2 \mid \mathcal{F}_t\right] \leq D_t^2$ and by induction $\mathbb{E}\left[D_{t+1}^2\right] \leq \mathbb{E}\left[D_0^2\right]$, which concludes the proof. $\square$

**Corollary A.1** (Convex smooth rate).
*If $\mathcal{L} : \mathbb{R}^d \to \mathbb{R}$ is convex and there exists $\theta^\star \in \mathbb{R}^d$ such that $\mathcal{L}(\theta^\star) = \mathcal{L}^\star$, and if for all $t \leq T$, it holds $\eta_t = \beta^{-1} \operatorname{Stab}(G_{t+1} \mid \mathcal{F}_t)$ (strong stability condition), and $\mathbb{V}\left[G_{t+1} \mid \mathcal{F}_t\right] \leq \alpha\|\nabla\mathcal{L}(\theta_t)\|_2^2 + \sigma^2$ (affinely bounded variance), then it holds*

$$\forall t \leq T, \quad \mathbb{E}\left[\mathcal{L}(\theta_{t+1})\right] - \mathcal{L}^\star \leq \Phi^{-1}\left(\Phi(\mathbb{E}\left[\mathcal{L}(\theta_0)\right] - \mathcal{L}^\star) + \frac{t}{2\beta}\right)$$

*where $\Phi : u \mapsto \dfrac{C^2\sigma^2}{3\,u^3} + (1 + \alpha)\dfrac{C}{2\,u}$ for $C = \mathbb{E}\left[\|\theta_0 - \theta^\star\|_2^2\right] \in \mathbb{R}_+$.*

Therefore, $T \geq \frac{2}{3}\beta C^2\sigma^2(\varepsilon^{-3} - \Delta_0^{-3}) + (1 + \alpha)\beta C(\varepsilon^{-1} - \Delta_0^{-1})$ implies $\mathbb{E}\left[\mathcal{L}(\theta_{T+1})\right] \leq \mathcal{L}^\star + \varepsilon$, which is a rate of $\mathcal{O}(T^{-1/3})$ if with additive noise $\sigma^2 > 0$, and $\mathcal{O}(T^{-1})$ in the case $\sigma^2 = 0$.

*Proof.* Define $C = \mathbb{E}\left[\|\theta_0 - \theta^\star\|_2^2\right]$ and $\varphi : u \mapsto u^2/C$. In order to use Lemma A.5, let us show that $\mathbb{E}\left[\|\nabla\mathcal{L}(\theta_t)\|_2^2\right] \geq \varphi(\mathbb{E}\left[\mathcal{L}(\theta_t) - \mathcal{L}^\star\right])$. By convexity of $\mathcal{L}$ and then by Cauchy-Schwarz inequality.

$$\mathcal{L}(\theta_t) - \mathcal{L}^\star \leq \nabla\mathcal{L}(\theta_t) \cdot (\theta_t - \theta^\star)$$

$$\mathbb{E}\left[\mathcal{L}(\theta_t) - \mathcal{L}^\star\right]^2 \leq \mathbb{E}\left[\|\nabla\mathcal{L}(\theta_t)\|_2^2\right] \cdot \mathbb{E}\left[\|\theta_t - \theta^\star\|_2^2\right]$$

Using additionally Lemma A.6 to get $\mathbb{E}\left[\|\theta_t - \theta^\star\|_2^2\right] \leq \mathbb{E}\left[\|\theta_0 - \theta^\star\|_2^2\right]$, this concludes the first part of the proof, that $\mathbb{E}\left[\|\nabla\mathcal{L}(\theta_t)\|_2^2\right] \geq \varphi(\mathbb{E}\left[\mathcal{L}(\theta_t) - \mathcal{L}^\star\right])$.

For the second part of the proof, apply Lemma A.5, with desingularizer $\Phi$ obtained by integration

$$d\Phi(u) = -\frac{\sigma^2 + (1+\alpha)\,\varphi(u)}{\varphi(u)^2} = -\frac{\sigma^2 + (1+\alpha)\,C^{-1}u^2}{C^{-2}u^4}$$

$$\Phi(u) = \frac{C^2\sigma^2}{3\,u^3} + (1+\alpha)\frac{C}{2\,u}$$

$\square$

**Bound inversion:** the condition to obtain $\mathbb{E}\left[\mathcal{L}(\theta_{T+1})\right] \leq \mathcal{L}^\star + \varepsilon$ with $T$ as a function of $\varepsilon$, i.e.

$$T_\varepsilon \geq \frac{2}{3}\beta C^2\sigma^2(\varepsilon^{-3} - \Delta_0^{-3}) + (1+\alpha)\beta C(\varepsilon^{-1} - \Delta_0^{-1})$$

can be rewritten with $\varepsilon$ as a function of $T$, as in the original statement of Corollary A.1, in the form

$$\varepsilon_T \leq \Phi^{-1}\left(\Phi\left(\Delta_0\right) + \frac{T}{2\beta}\right)$$

with $a = C^2\sigma^2/3$ and $b = (1+\alpha)C/2$ defining $\Phi(u) = au^{-3} + bu^{-1}$. This expression can be simplified at $y = \Phi(\Delta_0) + \frac{T}{2\beta}$ with the intermediate variables $p = -\frac{b^2}{3y^2}$ and $q = \frac{2b^3}{27y^3} + \frac{a}{y}$ using

$$\Phi^{-1}(y) = \sqrt[3]{\frac{q}{2} + \sqrt{\left(\frac{q}{2}\right)^2 + \left(\frac{p}{3}\right)^3}} + \sqrt[3]{\frac{q}{2} - \sqrt{\left(\frac{q}{2}\right)^2 + \left(\frac{p}{3}\right)^3}} + \frac{b}{3y} \qquad \text{(CVX-INV)}$$

This expression of $\varepsilon_T = \Phi^{-1}(y)$ is not any easier to use, hence our statement in the other $T_\varepsilon$ form.

**Corollary A.2** (Strongly-convex smooth rate).
*If $\mathcal{L}$ is $\mu$-strongly convex, and if for all $t \leq T$, it holds $\mathbb{V}\left[G_{t+1} \mid \mathcal{F}_t\right] \leq \alpha\|\nabla\mathcal{L}(\theta_t)\|_2^2 + \sigma^2$ (affinely bounded variance), and $\eta_t = \beta^{-1}\,\mathrm{Stab}\left(G_{t+1} \mid \mathcal{F}_t\right)$ (strong stability condition), then it holds*

$$\forall t \leq T, \quad \mathbb{E}\left[\mathcal{L}(\theta_{t+1})\right] - \mathcal{L}^\star \leq \Phi^{-1}\left(\Phi(\mathbb{E}\left[\mathcal{L}(\theta_0)\right] - \mathcal{L}^\star) + \frac{t}{2\beta}\right)$$

*where $\Phi : u \mapsto \dfrac{\sigma^2}{4\mu^2}\dfrac{1}{u} - \dfrac{1+\alpha}{2\mu}\log(u)$.*

For $T \geq \frac{\sigma^2\beta}{2\mu^2}(\varepsilon^{-1} - \Delta_0^{-1}) + (1+\alpha)\frac{\beta}{\mu}\log(\Delta_0/\varepsilon)$, where $\Delta_0 = \mathbb{E}\left[\mathcal{L}(\theta_0)\right] - \mathcal{L}^\star$, this implies that $\mathbb{E}\left[\mathcal{L}(\theta_{T+1})\right] - \mathcal{L}^\star \leq \varepsilon$. This is a rate of $\mathcal{O}(T^{-1})$ with additive noise $\sigma^2 > 0$ and a linear rate $\mathcal{O}(\exp(-\kappa T/(1+\alpha)))$ for $\kappa = \mu/\beta$ in the noiseless / multiplicative-noise case $\sigma^2 = 0$.

*Proof.* The proof is a straightforward application of Lemma A.5 with $\varphi : u \mapsto 2\mu u$, which satisfies $\mathbb{E}\left[\|\nabla\mathcal{L}(\theta_t)\|_2^2\right] \geq \varphi(\mathbb{E}\left[\mathcal{L}(\theta_t) - \mathcal{L}^\star\right])$, because by strong convexity of $\mathcal{L}$, it holds for all $x \in \mathbb{R}^d$ that $\|\nabla\mathcal{L}(x)\|_2^2 \geq 2\mu(\mathcal{L}(x) - \mathcal{L}^\star)$. It remains to compute the desingularizer $\Phi$ by integration

$$d\Phi(u) = -\frac{\sigma^2 + (1+\alpha)\,\varphi(u)}{\varphi(u)^2} = -\frac{\sigma^2 + 2(1+\alpha)\mu u}{4\mu^2 u^2}$$

$$\Phi(u) = \frac{\sigma^2}{4\mu^2}\frac{1}{u} - \frac{1+\alpha}{2\mu}\log(u)$$

$\square$

**Proposition A.3** (Non-convex rate).
*If for all $t \leq T$, it holds $\mathbb{V}\left[G_{t+1} \mid \mathcal{F}_t\right] \leq \alpha\|\nabla\mathcal{L}(\theta_t)\|_2^2 + \sigma^2$ (affinely bounded variance), and $\eta_t = \beta^{-1}\,\mathrm{Stab}\left(G_{t+1} \mid \mathcal{F}_t\right)$ (strong stability condition), then writing $\Delta_0 = \mathbb{E}\left[\mathcal{L}(\theta_0)\right] - \mathcal{L}^\star$, it holds*

$$\mathbb{E}\left[\frac{1}{T}\sum_{t<T}\|\nabla\mathcal{L}(\theta_t)\|_2^2\right] \leq (1+\alpha)\frac{2\beta\Delta_0}{T} + \sqrt{\frac{2\beta\Delta_0\sigma^2}{T}}$$

*Proof.* Starting from Lemma A.4 (valid by strong stability condition and affinely bounded variance),

$$\mathbb{E}\left[\mathcal{L}(\theta_{t+1}) \,|\, \mathcal{F}_t\right] - \mathcal{L}(\theta_t) \leq -\frac{1}{2\beta}\psi\left(\|\nabla\mathcal{L}(\theta_t)\|_2^2\right)$$

where $\psi : u \mapsto u^2 \cdot (\sigma^2 + (1+\alpha)\,u)^{-1}$ is a convex increasing function. Thus, taking total expectations and summing over iterates $t \in [T]$ to telescope $(a)$, and then using convexity of $\psi$ to bound $(b)$,

$$\psi\left(\mathbb{E}\left[\frac{1}{T}\sum_{t<T}\|\nabla\mathcal{L}(\theta_t)\|_2^2\right]\right) \underset{(b)}{\leq} \mathbb{E}\left[\frac{1}{T}\sum_{t<T}\psi\left(\|\nabla\mathcal{L}(\theta_t)\|_2^2\right)\right] \underset{(a)}{\leq} \frac{2\beta\Delta_0}{T}$$

where $\Delta_0 = \mathbb{E}\left[\mathcal{L}(\theta_0)\right] - \mathcal{L}^*$ is the expected initial optimization error. It remains to use the bound $\psi^{-1}(x) \leq (1+\alpha)x + \sigma\sqrt{x}$ (Lemma A.3), to obtain

$$\mathbb{E}\left[\frac{1}{T}\sum_{t<T}\|\nabla\mathcal{L}(\theta_t)\|^2\right] \leq \frac{2(1+\alpha)\beta\Delta_0}{T} + \sqrt{\frac{2\beta\Delta_0\sigma^2}{T}}.$$

$\square$

We thus recover the classical deterministic and stochastic regimes in, respectively, $O(1/T)$ and $O(1/\sqrt{T})$ depending on whether the additive variance term $\sigma^2$ is positive or equal to 0.

The same analysis would hold in a more general setting in which $\mathrm{Stab}\left(G_{t+1} \,|\, \mathcal{F}_t\right) \geq \varphi(\|\nabla\mathcal{L}(\theta_t)\|^2)$ and $x \mapsto x \cdot \varphi(x)$ is a positive, increasing and convex function.

## A.2 Estimation of stability ratio

**Lemma A.7.** *Let $B > 0$ and $(X_i)_{i\in[n]}$ be i.i.d. real random variables such that, for all $i \in [n]$, it holds $\mathbb{E}\left[X_i\right] = 0$ and $X_i \leq B$ almost surely. Then, for any $t > 0$, we have*

$$\mathbb{P}\left(\frac{1}{n}\sum_{i\in[n]}X_i \geq t\right) \leq \exp\left(-\frac{nt^2}{2B^2 f\left(\mathbb{V}[X_1]/B^2\right)}\right), \tag{1}$$

*where $f(x) = (1+x)^2/4$ if $x < 1$, and $f(x) = x$ otherwise. (In particular, $\forall x, f(x) \leq 1 + x$)*

*Proof.* Use Fan et al. [2015, Corollary 2.7] with $U_{i-1} = B$, note that $B^2 f(\mathbb{V}[X]/B^2) = C_{i-1}^2$ exactly matches the definition in the reference's notation, thus following the reference and simplifying constants $C_i$, we get for $v^2 = n\sum_{i=1}^{n} C_{i-1}^2 = nB^2 f(\mathbb{V}[X_i]/B^2)$, that it holds

$$\mathbb{P}\left(\sum_{i\in[n]}X_i \geq x\right) \leq \exp\left(-\frac{x^2}{2v^2}\right)$$

The result follows using $x = nt$. $\square$

**Lemma A.8.** *Let $(D_i)_{i\in[n]}$ be non-negative i.i.d. random variables with $\mathbb{E}\left[D_i^2\right] < +\infty$ and $\mathbb{E}\left[D_i\right] = D \in \mathbb{R}_+^*$. Then, for $\kappa = \mathbb{E}\left[D_i^2\right]/\mathbb{E}\left[D_i\right]^2 \in [1, \infty[$, it holds*

$$\mathbb{P}\left(\frac{1}{n}\sum_{i\in[n]}D_i \leq D/2\right) \leq \exp\left(-\frac{n}{8\kappa}\right)$$

*Proof.* Let $X_i = D - D_i$. Observe that $\mathbb{E}\left[X_i\right] = 0$, and $X_i \leq D$ almost surely. Additionally, by expanding the square, $\mathbb{V}[X_i] = \mathbb{E}\left[D_i^2\right] - D^2$.

Apply Lemma A.7 with $B = D$ and $t = D/2$ and use $f(x) \leq 1 + x$ to simplify the denominator with $D^2 f(\mathbb{V}[D_i]/D^2) \leq D^2 + \mathbb{V}[D_i] = \mathbb{E}\left[D_i^2\right]$. Therefore,

$$\mathbb{P}\left(\frac{1}{n}\sum_{i\in[n]}D_i \leq D/2\right) \leq \exp\left(-\frac{nD^2/4}{2\,\mathbb{E}\left[D_i^2\right]}\right) = \exp\left(-\frac{n}{8\,\mathbb{E}\left[D_i^2\right]/D^2}\right)$$

$\square$

**Lemma A.9** (Relative error of stability estimation). *Let $(X_i \in \mathbb{R}^d)_{i \in [n]}$ be iid random variables. Define $J = \frac{1}{n(n-1)} \sum_{i \neq j} X_i \cdot X_j \in \mathbb{R}$, $Z = \frac{1}{n} \sum_i \|X_i\|_2^2 \in \mathbb{R}_+$, then $S = \mathrm{clip}_{[0,1]}(J/Z) \in [0,1]$.*

*Write $\mu = \mathbb{E}[X] \in \mathbb{R}^d$ and $\sigma^2 = \mathbb{E}\left[\|X - \mu\|_2^2\right]$, and $\kappa = \mathbb{E}\left[\|X\|_2^4\right]/\sigma^4$. If $R = \|\mu\|_2^2/\sigma^2 \neq 0$, and if $n \geq 1 + a/R$ for a constant $a \geq 1$, then*

$$
\mathbb{E}\left[\left|\frac{S - R}{R}\right|^2\right] \leq \frac{48 + 4(\kappa - 1)}{a} + \frac{1}{R^2} \exp\left(-\frac{n}{8\kappa}\right)
$$

At $\kappa = 3$ (for a centered gaussian) and neglecting the fast-decreasing second term, this is a relative squared error of $56/a$, i.e. a relative error of order $7.48/\sqrt{a}$, which is below 1 as low as $a = 100$.

*Proof.* Let $N = \|\mu\|_2^2$ and $D = \mathbb{E}\left[\|X\|_2^2\right]$ be the numerator and denominator in $R = N/D$. Note that $\mathbb{E}[J] = N$ and $\mathbb{E}[Z] = D$. Proceed then by case disjunction: if on one hand $Z \leq D/2$, then $|S - R| \leq 1$ (both are in $[0, 1]$), while on the other hand if $Z \geq D/2$, then

$$
|S - R| \leq \left|\frac{J}{Z} - R\right| = \left|\frac{J - N}{Z} + N\left(\frac{1}{Z} - \frac{1}{D}\right)\right| \leq \frac{|J - N|}{Z} + \frac{N}{D}\frac{|Z - D|}{Z}
$$

$$
\leq \frac{|J - N|}{D/2} + \frac{N}{D}\frac{|Z - D|}{(D/2)} = 2R\frac{|J - N|}{N} + 2R\frac{|Z - D|}{D}
$$

Therefore joining both cases after taking squares,

$$
\left|\frac{S - R}{R}\right|^2 \leq 4\left|\frac{J - N}{N}\right|^2 + 4\left|\frac{Z - D}{D}\right|^2 + \frac{1}{R^2}\mathbb{1}\{Z \leq D/2\}
$$

Hence, after taking expectations and applying Lemma A.12 (numerator sample control) and Lemma A.11 (denominator variance), it holds for $n \geq 1 + a/R$ that

$$
\mathbb{E}\left[\left|\frac{S - R}{R}\right|^2\right] \leq 4\left(\frac{4}{a^2} + \frac{8}{a}\right) + 4\frac{\kappa - 1}{n} + \frac{1}{R^2}\mathbb{P}(Z \leq D/2)
$$

Additionally, by Lemma A.8, $\mathbb{P}(Z \leq D/2) \leq \exp\left(-\frac{n}{8s}\right)$ where $s = \mathbb{E}\left[\|X_i\|^4\right]/D^2 = \kappa$. Thus,

$$
\mathbb{E}\left[\left|\frac{S - R}{R}\right|^2\right] \leq 4\left(\frac{4}{a^2} + \frac{8}{a}\right) + 4\frac{R(\kappa - 1)}{a} + \frac{1}{R^2}\exp\left(-\frac{n}{8\kappa}\right)
$$

The result follows by using $a \geq 1$ and $R \leq 1$. $\qquad\square$

**Lemma A.10** (Uncentered kurtosis of isotropic normal distribution). *Let $X \in \mathbb{R}^d$ be a random variable with $X \sim \mathcal{N}(0, \sigma^2 I)$. It holds $\mathbb{E}\left[\|X\|_2^2\right] = d\sigma^2$ and $\mathbb{E}\left[\|X\|_2^4\right]/\mathbb{E}\left[\|X\|_2^2\right]^2 = \frac{d-1}{d} + \frac{3}{d}$*

*Proof.* By expanding the sum,

$$
\mathbb{E}\left[\|X\|_2^2\right] = \mathbb{E}\left[\sum_i X_i^2\right] = \sum_i \mathbb{E}\left[X_i^2\right] = d\sigma^2
$$

$$
\mathbb{E}\left[\|X\|_2^4\right] = \mathbb{E}\left[\left(\sum_i X_i^2\right)^2\right] = \sum_{i,j} \mathbb{E}\left[X_i^2 X_j^2\right]
$$

$$
= \sum_i \mathbb{E}\left[X_i^4\right] + \sum_{i \neq j} \mathbb{E}\left[X_i^2\right]\mathbb{E}\left[X_j^2\right] = d \cdot 3 \cdot \sigma^4 + d(d-1)\sigma^4
$$

The result follows by taking the quotient of both. $\qquad\square$

**Lemma A.11** (Kurtosis bound for the denominator).
*Let $(X_i \in \mathbb{R}^d)_{i \in [n]}$ be iid random variables with $\mathbb{E}\left[\|X\|_2^2\right] = Q$, and $Z = \frac{1}{n}\sum_{i \in [n]}\|X_i\|^2$. Then*

$$
\mathbb{E}\left[|Z - Q|^2\right] = \frac{1}{n}\left(\mathbb{E}\left[\|X\|_2^4\right] - \mathbb{E}\left[\|X\|_2^2\right]^2\right)
$$

*and thus for $\kappa = \frac{\mathbb{E}[\|X\|_2^4]}{\mathbb{E}[\|X\|_2^2]^2}$ (uncentered kurtosis of $X$), it holds $\mathbb{P}(|Z - Q| > \tau Q) \leq \frac{\kappa - 1}{n\,\tau^2}$*

Proof of the expectation is just expansion of the square and linearity of expectation. The second proposition is Chebyshev's inequality.

**Lemma A.12** (Numerator sample control).
*Let $(X_i \in \mathbb{R}^d)_{i \in [n]}$ be iid random variables with $\mathbb{E}[X] = \mu$, and $J = \frac{1}{n(n-1)} \sum_{i \neq j} X_i \cdot X_j$. If $\mu \neq 0$ and if $n \geq 1 + c \cdot \mathbb{E}[\|X - \mu\|_2^2] / \|\mu\|_2^2$ then it holds*

$$\frac{\mathbb{E}\left[\left|J - \|\mu\|_2^2\right|^2\right]}{\|\mu\|_2^4} \leq \frac{4}{c^2} + \frac{8}{c}$$

This is an immediate corollary of the following lemma.

**Lemma A.13** (Variance bound for the Jackknife numerator).
*Let $(X_i \in \mathbb{R}^d)_{i \in [n]}$ be iid random variables with $\mathbb{E}[X] = \mu \in \mathbb{R}^d$, and $J = \frac{1}{n(n-1)} \sum_{i \neq j} X_i \cdot X_j$. Then it holds $\mathbb{E}[J] = \|\mu\|_2^2$, and*

$$\mathbb{E}\left[\left|J - \|\mu\|_2^2\right|^2\right] \leq 4\frac{\mathbb{E}\left[\|X - \mu\|_2^2\right]^2}{n(n-1)} + \frac{8}{n} \mathbb{E}\left[\|X - \mu\|_2^2\right] \cdot \|\mu\|_2^2$$

*Proof.*

$$J - \mathbb{E}[J] = \frac{1}{n(n-1)} \sum_{i \neq j} \left(X_i \cdot X_j - \mu^2\right)$$

$$= \frac{1}{n(n-1)} \sum_{i \neq j} \left((X_i - \mu) \cdot (X_j - \mu) + (X_i + X_j)\mu - 2\mu^2\right)$$

$$= \frac{1}{n(n-1)} \left(\left(\sum_{i \neq j}(X_i - \mu) \cdot (X_j - \mu)\right) + \left(2(n-1)\sum_i X_i \cdot \mu\right) - 2n(n-1)\mu^2\right)$$

$$= \frac{1}{n(n-1)} \left(\sum_{i \neq j}(X_i - \mu) \cdot (X_j - \mu)\right) + 2\left(\frac{1}{n}\sum_i X_i - \mu\right) \cdot \mu$$

$$:= A + B$$

As a sanity check, observe that $\mathbb{E}[J - \mathbb{E}[J]] = 0$ because $\mathbb{E}[A] = 0$ and $\mathbb{E}[B] = 0$.

We will use the (crude) bound $\mathbb{E}\left[|J - \mu^2|^2\right] \leq 2\mathbb{E}[A^2] + 2\mathbb{E}[B^2]$. Let us compute each.

$$\mathbb{E}[B^2] = 4\mathbb{E}\left[\left(\left(\frac{1}{n}\sum_i X_i - \mu\right) \cdot \mu\right)^2\right] \leq 4\mathbb{E}\left[\left\|\frac{1}{n}\sum_i X_i - \mu\right\|_2^2\right] \cdot \mu^2 \leq 4\frac{\sigma^2}{n}\mu^2$$

On the other hand, by Lemma A.14, $\mathbb{E}[A^2] \leq 2\mathbb{E}\left[\|X - \mu\|_2^2\right]^2 / (n(n-1))$. Thus the conclusion,

$$\mathbb{E}\left[\left|J - \|\mu\|_2^2\right|^2\right] \leq 2\mathbb{E}[A^2] + 2\mathbb{E}[B^2] \leq 4\frac{\mathbb{E}\left[\|X - \mu\|_2^2\right]^2}{n(n-1)} + \frac{8}{n}\mathbb{E}\left[\|X - \mu\|_2^2\right] \cdot \|\mu\|_2^2$$

$\square$

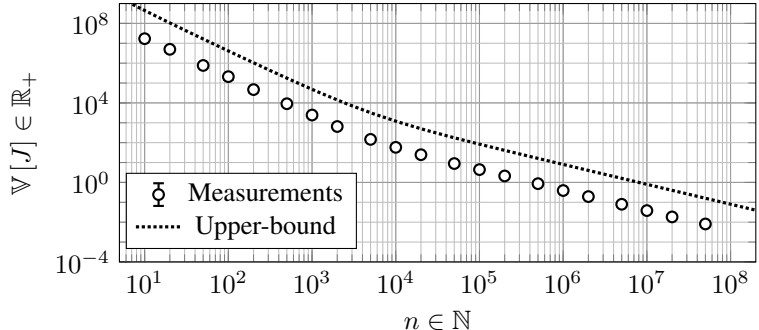

Figure 10: Empirical measurements of $\mathbb{E}\left[\,|J_n - \mathbb{E}\left[\,J_n\,\right]|^2\,\right]$ as a function of $n$ (mean and 5-sigma confidence interval for the mean, $10^3$ samples) vs Lemma A.13 upper-bound, for isotropic gaussians in dimension $d = 10$ with noise $\sigma_0 = 10^2$ per coordinate, thus $\sigma^2 = d\,\sigma_0^2 = 10^5$, and $\|\mu\|_2^2 = d$.

**Lemma A.14** (Variance of the squared-mean U-statistic).
*Let $(C_i \in \mathbb{R}^d)_{i \in [n]}$ be iid random variables with $\mathbb{E}\left[\,C_i\,\right] = 0$ and $\mathbb{E}\left[\,\|C_i\|_2^2\,\right] = \sigma^2 \in \mathbb{R}_+$. Define $A = \frac{1}{n(n-1)} \sum_{i \neq j} C_i \cdot C_j$. Then $\mathbb{E}\left[\,A^2\,\right] \leq 2\sigma^4 / (n(n-1))$.*

This is the usual analysis of variance of a U-statistic by intersection disjunction, see for instance the lecture notes Jordan [2007] for Berkeley's Stat 210B, or the more conventional reference *Asymptotic Statistics* [Vaart, 1998]. An empirical verification and tightness evaluation is performed in Figure 11.

*Proof.* Starting from the definition of $A$

$$A^2 = \frac{1}{n^2(n-1)^2} \sum_{i \neq j} \sum_{k \neq l} (C_i \cdot C_j)(C_k \cdot C_l)$$

Proceed by case disjuction:

- if $\{i,j\} \cap \{k,l\} = \varnothing$, then $\mathbb{E}\left[\,(C_i \cdot C_j)(C_k \cdot C_l)\,\right] = \mathbb{E}\left[\,C_i \cdot C_j\,\right]\mathbb{E}\left[\,C_k \cdot C_l\,\right] = 0$.

- if $\#(\{i,j\} \cap \{k,l\}) = 2$, then $\mathbb{E}\left[\,(C_i \cdot C_j)(C_k \cdot C_l)\,\right] = \mathbb{E}\left[\,(C_i \cdot C_j)^2\,\right]$, and by Cauchy-Schwarz inequality, it holds $\mathbb{E}\left[\,(C_i \cdot C_j)^2\,\right] \leq \mathbb{E}\left[\,C_i^2 C_j^2\,\right] \leq \mathbb{E}\left[\,C_i^2\,\right]\mathbb{E}\left[\,C_j^2\,\right] \leq \sigma^4$.

- if $\#(\{i,j\} \cap \{k,l\}) = 1$, then without loss of generality $i = k$ and $j \neq l$. Therefore
  $$\mathbb{E}\left[\,(C_i \cdot C_j)(C_k \cdot C_l)\,\right] = \mathbb{E}\left[\,((C_i \cdot C_l)C_i) \cdot C_j\,\right] = \mathbb{E}\left[\,(C_i \cdot C_l)C_i\,\right] \cdot \mathbb{E}\left[\,C_j\,\right] = 0$$

It remains to take expectations and count the number of size-2 intersections.

$$\mathbb{E}\left[\,A^2\,\right] \leq \frac{1}{n^2(n-1)^2} \sum_{i \neq j} 2\sigma^4 \leq \frac{2\sigma^4}{n(n-1)}$$

$\square$

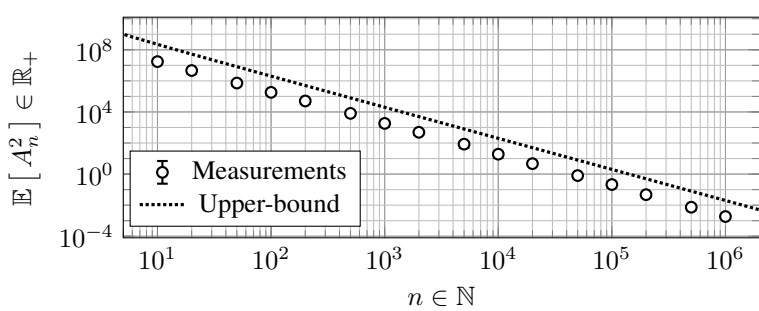

Figure 11: Empirical measurements of $\mathbb{E}\left[\,A_n^2\,\right]$ as a function of $n$ (mean and 5-sigma confidence interval for the mean, 500 samples) versus upper-bound used in Lemma A.14, for isotropic gaussians in dimension $d = 10$ with noise $\sigma_0 = 100$ per coordinate, thus $\mathbb{E}\left[\,\|C_i\|_2^2\,\right] = \sigma^2 = d\,\sigma_0^2 = 10^5$.

# B Influence of learning rate parameters ($\eta$-scan) on Problem QWC

We present results of all algorithms on Problem QWC at various noise levels, for all learning rate parameters tried in our experimental protocol. Flatter lines indicate less sensibility to the hyperparameter, aligned minima indicate ability to tune on short horizons. The smooth standard limit step $\beta^{-1} = 1$ is displayed as a vertical dotted line, for all algorithms with smooth claims.

## B.1 SGD with constant, scheduled, or stability-adjusted learning rates

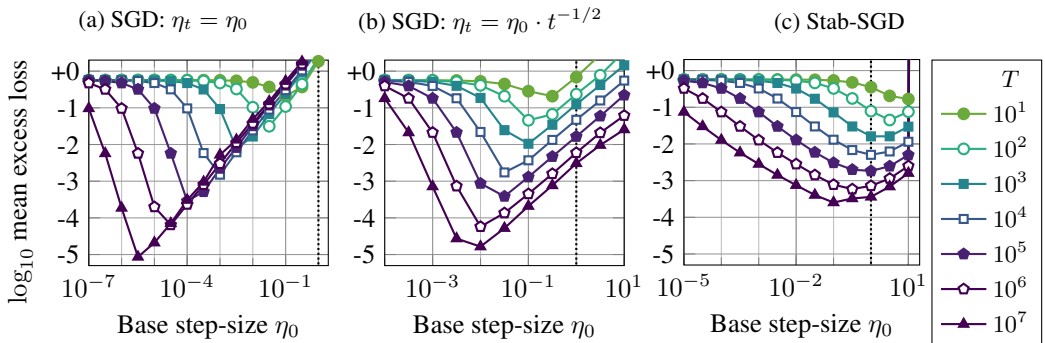

Figure 12: Problem QWC with additive gaussian noise of variance $\sigma^2 = d$. Excess loss versus base learning rate $\eta_0 \in \mathbb{R}_+^*$ and training time $T \in \mathbb{N}$.

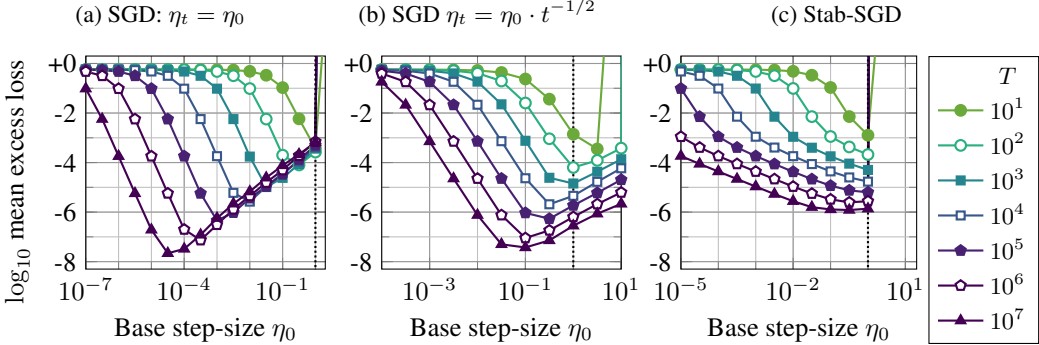

Figure 13: Problem QWC with additive gaussian noise of variance $\sigma^2 = 10^{-4} \cdot d$. Excess loss versus base learning rate $\eta_0 \in \mathbb{R}_+^*$ and training time $T \in \mathbb{N}$.

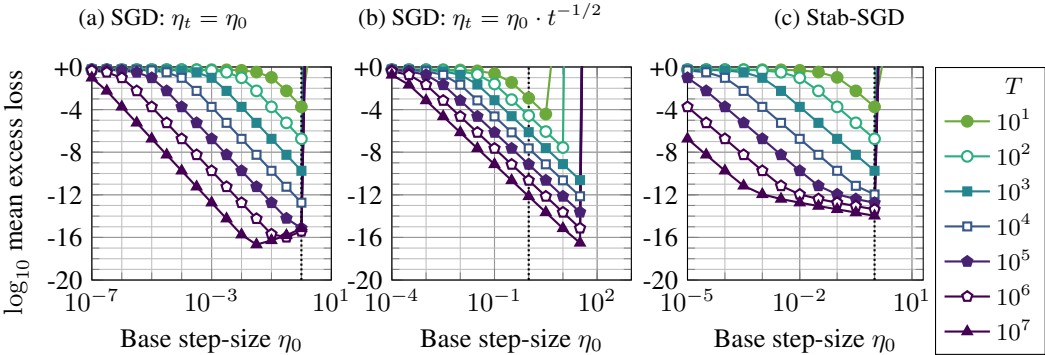

Figure 14: Problem QWC with additive gaussian noise of variance $\sigma^2 = 10^{-16} \cdot d$. Excess loss versus base learning rate $\eta_0 \in \mathbb{R}_+^*$ and training time $T \in \mathbb{N}$.

Fig. 12, Fig. 13 and Fig. 14 show evolution of the mean excess loss as a function of the base learning rate $\eta_0$ (before applying any scheduler) and the total training time $T \in \mathbb{N}$ (a.k.a. "horizon").

The dependence of the optimal base learning rate on the horizon $T$ is visible for both SGD with constant learning rate and with $t^{-1/2}$ schedule. Additionally, these optimal base learning rates are seen to shift between the two figures, when the noise levels vary. In particular, this means that the learning rate of mini-batch SGD must be re-tuned if the batch size (i.e. noise level) is altered.

Consistently with other experiments, the optimal learning rates for long horizons ($T \geq 10^7$) are associated with a long plateau at initialization. This implies that models tuned for long horizons are essentially unusable at mid-training (no better than initialization), thus it is meaningless to consider an "optimal trajectory", or a horizon-independent "optimal learning rate"; on the contrary, the horizon plays a central role in evaluating the quality of the model. This effect is much less pronounced with Stab-SGD, with little to no movement around the prescribed rate $\eta_0 = \beta = 10^0$ across noise levels.

## B.2 D-Adapt

We repeat the experiment at multiple noise levels with the D-adapt algorithm, Defazio and Mishchenko [2023, Algorithm 2]. We run the experiment with the hyperparameters $D = 2$ and $D = 200$ separately, and sweep over all "learning rates" $G^{-1}$ for each case.

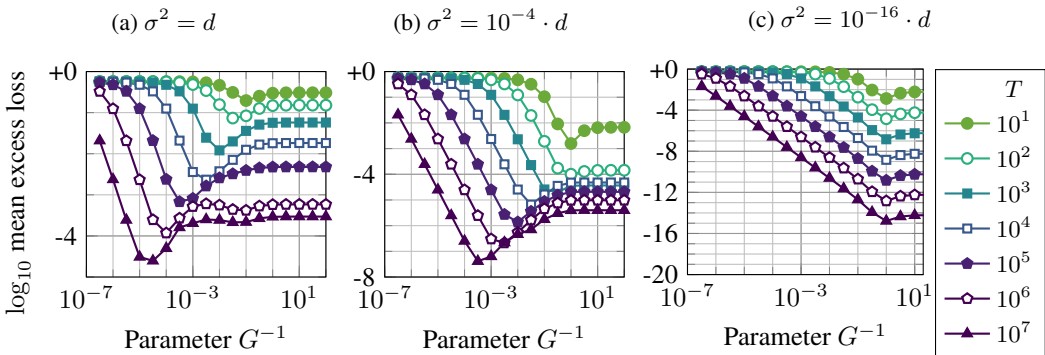

Figure 15: Performance of D-adapt algorithm, for $D=2$, on Problem QWC at various noise levels.

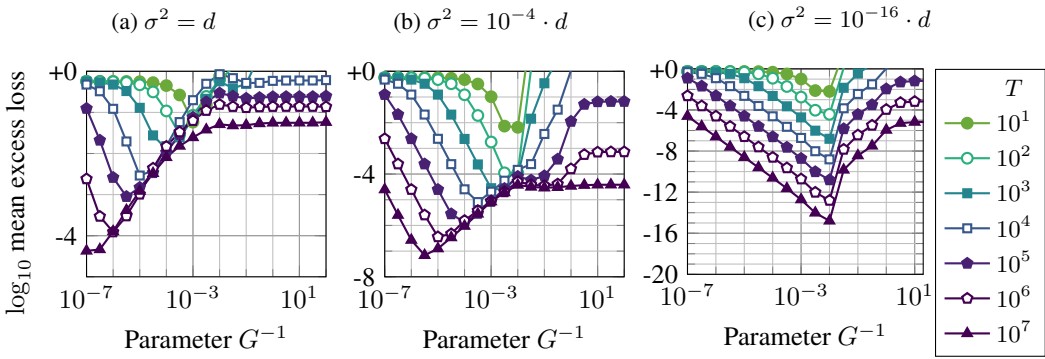

Figure 16: Performance of D-adapt algorithm, for $D=200$, on Problem QWC at various noise levels.

## B.3 "Schedule-free SGD"

We repeat the experiments with the "Schedule-free SGD" algorithm from "The Road Less Scheduled", as it is described in the main text: Defazio et al. [2024, Sec. 2, Eq 3-5], i.e. with hyperparameters $\beta = 0.9$ and $x$-step schedule $c_t = 1/(t+1)$ as prescribed in Sec. 2 §2.

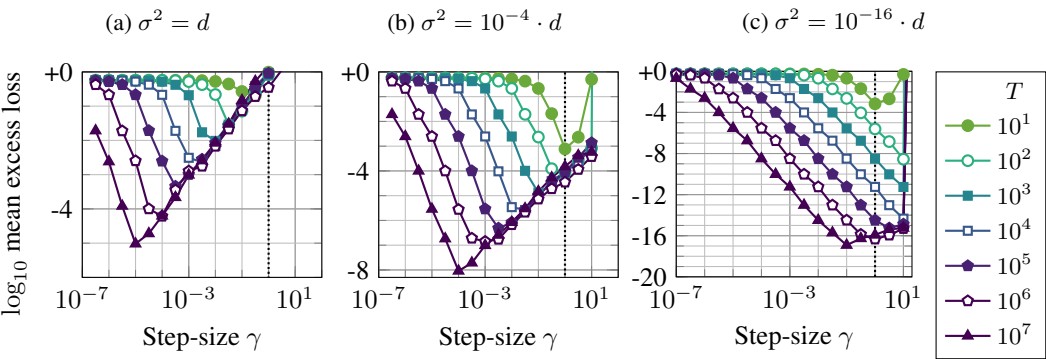

Figure 17: Performance of the "Schedule-free SGD" algorithm, on Problem QWC at various noise levels. We observe saturation at all noise levels, this is inconsistent with the idea that this algorithm can be used instead of a scheduler for SGD.

To contrast this with the theoretical predictions in the reference, note that Defazio et al. [2024, Thm 1] only gives convergence (in the Lipschitz model) with the horizon-dependent hyperparameter $\gamma = D\,T^{-1/2}$. The smooth result Defazio et al. [2024, Appendix Corollary 2] uses a time-varying parameter $\beta_t$, such as $\beta_t = 1/(5(t+1))$ (obtained by injecting the bounds on $w_t$ and $\alpha_t$ of Corollary 2 into their definition in Thm 5), to guarantee speed $\mathcal{O}(D^2\beta/T^2 + D\sigma/\sqrt{T})$, and uses an "optimistic online learning algorithm" for $z$ – the one given in appendix Sec D.1 uses a vanishing learning rate.

### B.4 COCOB - Coin-betting approach

We perform the same experiments with the Continuous Coin-Betting algorithm (COCOB) Orabona and Tommasi [2017, Algorithm 1], designed for the setting of convex online learning with Lipschitz losses and almost surely bounded gradients. Although this experiment uses smooth losses with gaussian noise (unbounded with finite variance), the performance of both this algorithm and its "Backprop" version more adapted to the non-convex setting remain competitive.

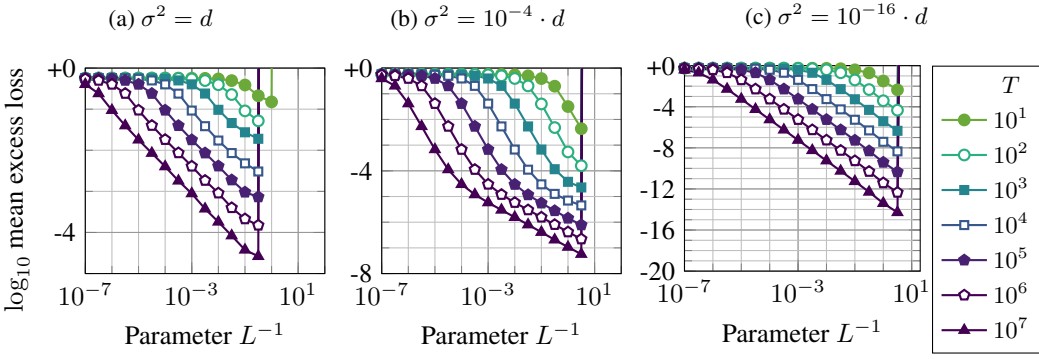

Figure 18: Performance of the "COCOB" algorithm, on Problem QWC at various noise levels. The algorithm uses a hyperparameter $(L_i)_i \in \mathbb{R}^d_+$, which we set identically for all directions for this experiment, this being the only reasonable choice without a canonical basis.

As observed in Fig. 18, the alignement of the optimal hyperparameter across training horizons is excellent, despite the mismatch in settings (Lipschitz objective in the theory, versus quadratic loss in the experiment, which is uniformly 1-smooth but not Lipschitz on the entire domain). The value of the limit learning rate however is perhaps not so intuitive, since it is no longer directly linked to $\beta^{-1}$.

Fig. 19 shows the results of COCOB-Backprop Orabona and Tommasi [2017, Algorithm 2].

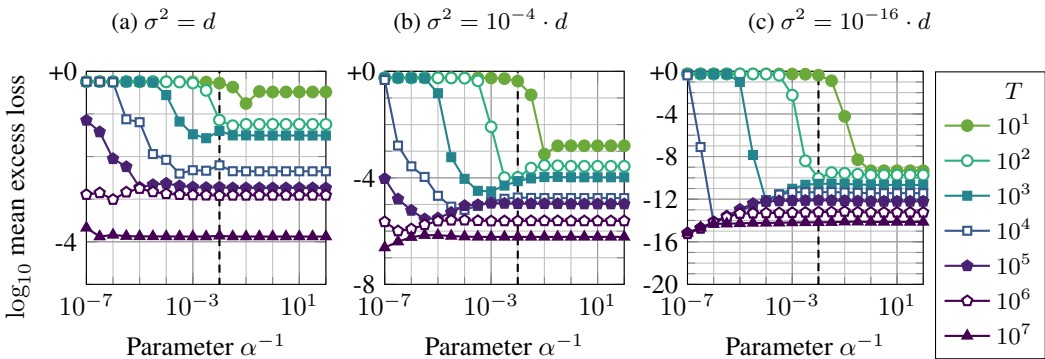

Figure 19: Performance of the "COCOB-Backprop" algorithm, on Problem QWC at various noise levels. The vertical line depicts the default value ($\alpha = 10^{+2}$) suggested to make this algorithm completely "parameter-free" (in the sense that is has no parameters to tune).

The sensitivity to the hyperparameter is essentially non-existant except near the initialization. The performance does not quite match that of SGD. For instance at $\sigma^2 = 10^{-16} \cdot d$, SGD (both constant-step and $t^{-1/2}$-scheduled) reach $10^{-17}$ after $10^7$ iterations (cf. Figure 14), while COCOB-Backprop reaches only $10^{-15}$. The observation of such a gap on a single problem does not allow general conclusions on the behavior of the algorithm (usually evaluated only in worst-case performance) but remains marginally informative. The gap in performance was most apparent on Problem QSC.

## C   ResNet Training Experiments

**Methods (additional details).** Consistently with the original experimental protocol He et al. [2015a, Section 3.4], we use the initialization taken from He et al. [2015b], also known as "Kaiming" initialization. This explains in particular the large initial loss, due to large values in the last layer at initialization under such scheme. Since the number of samples is not perfectly divisible by the batch size, our last batch in each epoch is smaller, we do not use a multiplicative correction for this altered size. We present in the following pictures results over 20 random seeds. Since one in those twenty essentially failed to train (loss nearly stalled at initial value), we present median and quartiles for error bars instead of means, which are less sensitive to large but rare values.

### C.1   Loss and accuracy across multiple runs (full scale)

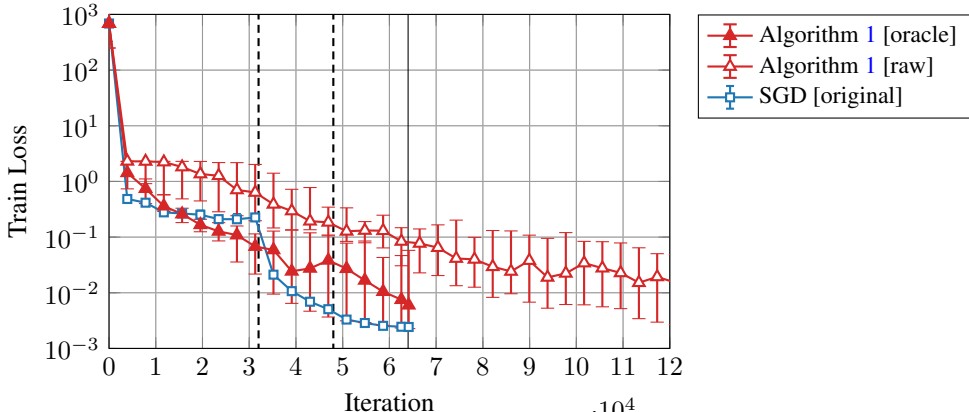

Figure 20: median (and quartiles as error bars) of the training loss as a function of iterations.

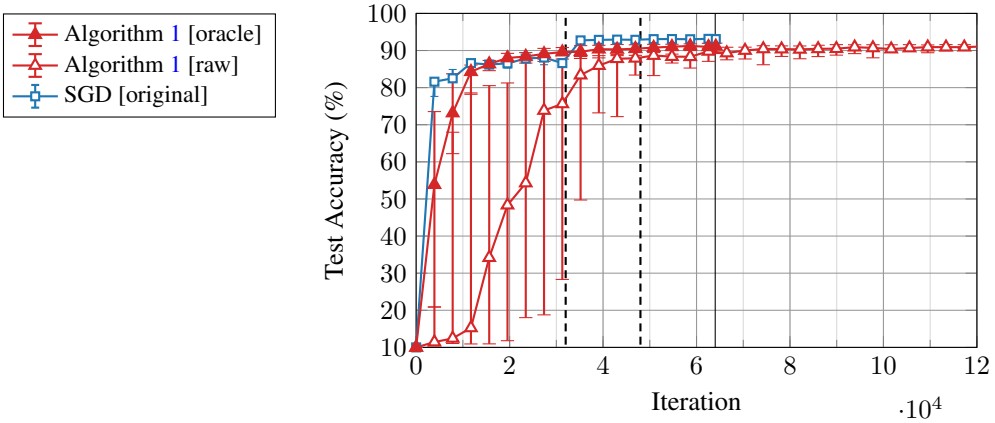

Figure 21: Median (and quartiles as error bars) of the test accuracy as a function of iterations.

## C.2 Stability ratio along trajectory, and kurtosis estimations

Fig. 22 shows the Stability Ratio and estimated kurtosis of gradients along the trajectory. Except for one run with very high kurtosis (> 40), all observed values are below 10 for most of the trajectory, leading to an error of $44 + 4\kappa \leq 84$ (Lemma 1) which is below our choice of $\zeta = 100$.

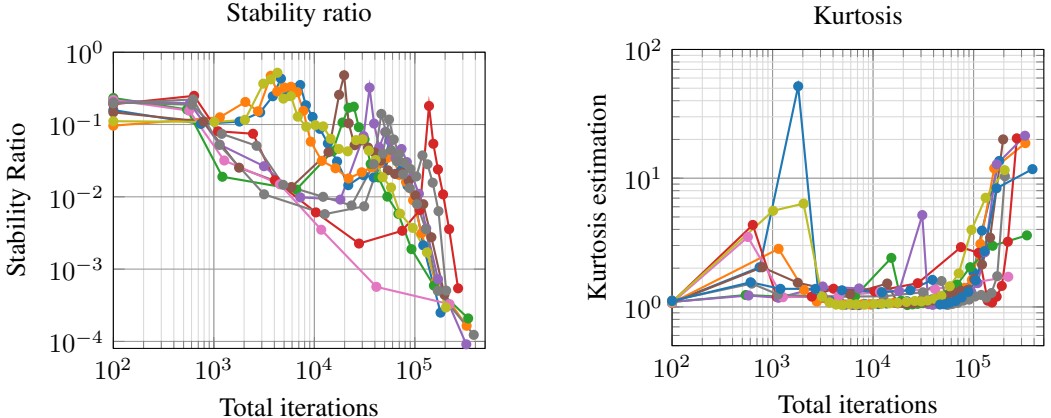

Figure 22: Stability ratio and kurtosis along trajectory (10 random seeds).

