# OpenReview forum: "Stab-SGD: Noise-Adaptivity in Smooth Optimization with Stability Ratios"
_NeurIPS.cc/2025/Conference — NeurIPS 2025 poster_

### Official Review · Reviewer_aGBz · 2025-06-17

**Clarity:** 3
**Significance:** 2
**Originality:** 3
**Rating:** 3
**Confidence:** 4

**Summary:**

This work considers hyperparameter-free stochastic optimization for smooth loss functions. The authors propose to use the signal-to-noise ratio for the gradient oracle to derive an appropriate adaptive learning rate that ensures a decay in the loss function in expectation. Under the assumption that the so-called stability ratio is known at each point, the authors give convergence rates for $\beta$-smooth loss functions that are convex, strongly convex and non-convex, respectively.  The derived rates are comparable to those found for SGD with decaying learning rate schedules. Numerical experiments are presented that compare the performance of the proposed variant of SGD to other schedule-free algorithms.

**Questions:**

- In the convex setting, the convergence rate derived in this work matches the rate given in Bach and Moulines (2011). In line 124, the authors claim that this rate is conjectured to be the optimal horizon-free last-iterate rate for SGD with schedule $\eta_t=\eta_0 t^\kappa$. However,  Theorem~3.1 in Liu,Zhou (2024), Revisiting the last-iterate Convergence of Stochastic Gradient Methods, appears to improve this rate to $O(\log(T)/\sqrt T)$.  Could the authors clarify their statement?
- I am wondering why the bounds in Theorem 1 and Theorem 2 are presented in a different way compared to the bounds in Theorem 3. As far as I understand, since the learning rate does not depend on the time horizon, it would be possible to give an upper bound for the optimality gap in terms of the current iteration number $T$, as it has been done in Theorem 3.
- It would be interesting to analyze how using the same gradient estimates for both the learning rate and the descent direction would change the dynamics of Stab-SGD. Can the convergence results be recovered in these settings with dependencies between the learning rate and the gradient estimate?

**Ethical Concerns:**

["NO or VERY MINOR ethics concerns only"]

**Final Justification:**

Overall, I agree with reviewer 2Qgz in his main criticism. Firstly, there is a large gap between the algorithm for which this paper derived convergence rates (using the stability ratio) and Algorithm 1.

If the stability ratio is updated in every iteration, the new theoretical statement the authors proposed in response to my comments (convergence rates with high probability if the stability ratio is approximated well enough) together with the approximation result in Lemma 1 is a step in the right direction. However, we are not able to verify the details of the proposed statement and the corresponding proof.

Moreover, the general form of Algorithm 1 does introduce new hyperparameters, which is counterproductive to the overall aim of the work.

The use and analysis of stochastic step-sizes itself is not a novelty or a major technical difficulty. It is standard practice e.g. in Reinforcement Learning.

Even taken into account the comments of reviewer SUNZ and the authors regarding hyerparameter training budget, I still believe the numerical experiments are not convincing especially compared to the author parameter-free algorithms.

Therefore, I believe the paper has two main contributions:
- It starts a discussion on noise-adaptive learning rates.
- It presents an algorithm (when having access to the stability ratio) that adaptively chooses a constant learning rate for situations with asymptotically degenerate noise and a decaying learning rate for situations with noise.

I think this is a strong idea, but I am hesitant to recommend acceptance due to the limitations of the work. Therefore, I am keeping my score.

**Limitations:**

Yes.

**Paper Formatting Concerns:**

There are no paper formatting concerns.

**Quality:**

3

**Strengths And Weaknesses:**

**Strengths:**
The paper is written in a clear and concise way. Adapting the learning rate to the noise intensity is a nice idea which allows to present a framework for convergence of SGD without using a decaying learning rate schedule. In fact, the proposed learning rates adapt to the asymptotic noise regime. For vanishing noise at the minimum the learning rate does not vanish in the limit, allowing Stab-SGD to achieve linear convergence. For non-vanishing noise the stability ratio and, thus, the learning rate converges to $0$, recovering the $O(T^{-1})$ convergence rates for SGD with decaying learning rate schedule. \

**Weaknesses:**
The stability ratio is a theoretical object and has to be approximated in practice using Monte Carlo simulations, requiring more gradient oracle queries. It is not clear how the estimation of the stability ratio and the convergence of the Stab-SGD algorithm interact, since the theoretical guarantees are presented under the assumption that the exact stability ratio is used for the learning rates. Moreover, it seems inefficient to generate estimates of the gradient for adapting the learning rate and then sample new, independent gradient estimates for determining the descent direction. The numerical experiments are not convincing, so that the contribution of this work is rather theoretical than practical.


Some minor remarks are listed below:
- Line 75: It should be assumed that $G$ is unbiased.
- Lemma A.6: Convexity of $\mathcal L$ should be stated as an assumption in the lemma.
- Corollary A.1: The definition of $\Delta_0$ should be recalled.

---

> ### Author Rebuttal · Authors · 2025-07-30
>
> Thank you for your time reviewing this submission and for the excellent reference that we had missed.
> We have updated the corresponding literature review to reflect the new state of non-adaptive last-iterate
> rates. Indeed, by using not just the power scheduler $\eta_t = \eta_0 t^\alpha$ but instead
> a capped $\eta_t = \min(1/\beta, \eta_0 t^\alpha)$, the authors break past the conjectured $T^{-1/3}$ rate.
> Unfortunately, the two rates presented in the weakly and strongly convex case do not use directly coinciding
> step-sizes, so there is little hope of re-using the analysis tools presented in this adaptive case.
> Therefore, the task of adaptively matching their improved $\log(T) / \sqrt{T}$ last-iterate rate
> while maintaining fast convergence on strongly convex problems will remain an open problem for the time being.
>
> As you have noted, the new theoretical framework we present allows handling stochastic step-sizes
> (because $\eta_{t+1}$ depends on $X_t$ which is random), but leaves unanswered the question of
> interactions between the ratio estimation and choice of direction.
> For the practical variant (Alg 1) we have chosen to discard gradient samples used for the stability estimation,
> precisely to minimize this interaction, we thus do not consider this an inefficiency but a safety measure.
> This could be improved by future works, for instance by demonstrating (by a new analysis) that the interaction
> between these two components can be controlled.
> We believe that the new techniques presented here to handle the stochastic step-size in a noise-adaptive manner
> will be a strong building block for future theories tackling these interactions, justifying the publication
> of these proof techniques and the possibility of noise-adaptivity that they demonstrate.
> Note as well that if the stability estimation is seldom updated, then the effect of discarding these samples
> for the choice of descent direction is negligible.
>
> We have updated the appendix to reflect both versions $T \leq f(\varepsilon)$ and $\varepsilon \leq g(T)$
> for theorems 1, 2, and 3. As you observed, they are strictly equivalent, and choosing one or the other is only a matter
> of whether one prefers using the simple function $f(u) = a u^{-3} + b u^{-1}$
> or the less easily-expressed $f^{-1}$ function in the bound.
> We have also fixed the missing clarifications on unbiased-ness, use of $\Delta$ and convexity rightfully noted.

---

> > ### Comment · Reviewer_aGBz · 2025-08-04
> > **Thank you for your rebuttal**
> >
> > I want to thank the authors for their responses and I appreciate the minor changes to the manuscript. After reading the reviews and the subsequent discussion my evaluation remains unchanged. Introducing the stability ratio in order to cover the SGD case with non-degenerate noise (and vanishing learning rates) and the degenerate noise case (with constant learning rates) in one algorithm is a nice idea. At least from a theoretical point of view, it starts an interesting discussion on noise-adaptive learning rates and poses several open question worth pursuing in the future.
> > However, the presented theoretical analysis is limited to the case where the stability ratio is known (which simplifies the convergence analysis significantly). Furthermore, the presented numerical results raise questions on the practical relevance of the proposed idea.

---

> > > ### Author Response · Authors · 2025-08-05
> > >
> > > We acknowledge that our contribution is mainly theoretical, designed to advance the
> > > state of machine learning theory in the direction of noise-adaptive last-iterate
> > > algorithms better suited to the use in deep learning. We consider these advances
> > > novel, making use of new analysis techniques, and a good basis for discussion of
> > > future advances in this direction in the community.
> > >
> > > Nonetheless, we also added a practical version (Alg 1) which does not use
> > > the stability oracle, and instead relies on gradient samples only. We presented
> > > a fair evaluation with ResNets and a comparable number of gradient queries,
> > > which we consider to be promising, and therefore support the high significance
> > > of these theoretical developments.
> > > As discussed with Reviewer SUNZ ($\S2$), these results have to be interpreted
> > > with care, because the aim of adaptive algorithms is to reduce the hyperparameter-tuning
> > > cost, not the training cost itself.
> > > Therefore, with a training run of $T$ iterations, and $k$ runs to tune the scheduler
> > > (with $k$ of order 10 to 100, for a total cost of $k \times T$), the observed identical
> > > performance after $2 T$ iterations (fair because they use the same number of gradient queries) is promising as soon as $k > 2$.
> > >
> > > Regarding the question of applicability of the analysis to the Alg 1 practical variant
> > > of the Stab-SGD algorithm, we are now able to add the following claim to our analysis:
> > > if the stability ratio $\operatorname{Stab}(G_k)$ is estimated as $S_k$ with $S_k / \operatorname{Stab}(G_k) \in [1-\varepsilon, 1 + \varepsilon]$ with high probability, then we can ensure that there is a high probability event $Z$ such that $\mathbb{E}[f(X_k) \mid Z] \leq h(k)$ for the same function $h$ as in our analysis yielding $\mathbb{E}[f(X_k)] \leq h(k)$ for the oracle-using Stab-SGD. We have already addressed how to construct estimators yielding accurate estimations with high-probability in Lemma 1, provided only that the kurtosis of gradient samples is finite.
> > > This estimation accuracy can even be strengthened under stronger hypotheses,
> > > such as sub-gaussianity of gradient samples.
> > > In the absence of a consensus in the community on which hypothesis should be chosen to state that
> > > gradient variance is a quantity that can be accurately estimated from samples (e.g. because gradients are not too heavy-tailed),
> > > we have refrained from adding unnecessarily long discussions on the various such
> > > choices of characterizations.
> > > We hope this is more convincing evidence of the relevance of this analysis in finer models of practical deployment.

---

> > > > ### Comment · Reviewer_aGBz · 2025-08-08
> > > >
> > > > Thank you for your response. I acknowledge that the presented analysis is a good basis for further discussion in the direction of noise-adaptive learning rates. That said, the empirical evaluation and the theoretical analysis remains limited.
> > > >
> > > > I agree that the new statement would be a good step towards providing a theoretical foundation for Algorithm 1.

---

### Official Review · Reviewer_hYFS · 2025-06-23

**Clarity:** 3
**Significance:** 3
**Originality:** 2
**Rating:** 5
**Confidence:** 2

**Summary:**

This paper proposes modulating step sizes with a measure of signal-to-noise of the gradient.

**Questions:**

N/A

**Ethical Concerns:**

["NO or VERY MINOR ethics concerns only"]

**Final Justification:**

Based on other reviews and rebuttals, I standby my assessment.

**Quality:**

2

**Strengths And Weaknesses:**

This paper has an interesting ideas. I like the comparison against various other methods in the last-iterate analysis setting. The use of online estimates of the stability ratio seems sensible, particularly the jackknife estimate.

- The stability ratio appears to be a signal-to-noise estimate. Some discussion of this would be helpful. Is there existing literature using SNR estimates to correct the step size? I did some quick googling and found the following two papers, which are related but not an exact overlap:
https://openreview.net/forum?id=TKXMPtCniG&noteId=utYnKNrXWv
https://www.sciencedirect.com/science/article/abs/pii/S0167865524002629

- I appreciate the large set of methods compared against in the experiment section, it's good to see a comprehensive algorithmic comparison.
- Line 39 "On the contrary, there is growing evidence that such aggregation is unnecessary": I wouldn't really agree with this statement. The Schedule-Free algorithm that is shown to work very well in this papers experiment section actually does use averaging, together with an additional layer of momentum that fixes the poor performance of standard Polyak averaging. The theory of "Optimal Linear Decay Learning Rate Schedules" (Defazio et. al. 2023) is a good counter-point though, showing that in some cases averaging can be replaced with linear decay schedules without sacrificing convergence rate guarantees. Perhaps this statement could be expanded and qualified?
- The experiments suggest that estimating the stability ratio well is difficult particularly as you approach the optimum. I think this is the main limitation of this line of work.
- I don't understand the D-Adapt results, doesn't it set D automatically, so what are the indicated D values in the legend?
- Theory results are clear and well-stated, and placed within context.

---

> ### Author Rebuttal · Authors · 2025-07-30
>
> Thank you for your time reviewing this submission.
> Indeed, the stability ratio is directly related to a signal-to-noise ratio, but remains well defined
> and finite when the noise vanishes, which eases all statements of convergence.
> The idea of reweighting learning rates using noise estimations seems intuitive, but underdeveloped
> in the community, possibly because getting a provably-good reweighting comes with many theoretical challenges.
> Previously,
> Schaul \& Zhang \& LeCun (2012) "No more pesky learning rates" proposed
> to use hessian estimates reweighted by a coordinate-specific noise estimation,
> but the algorithm did not gain traction in the following decade, and the original
> work proposes no convergence analysis.
> The first reference (Guo-Qing, Jinlong , Zixiang, Lin, Wei [2023])
> uses a signal-to-noise ratio $\lVert \mu_i \rVert_2^2 / \sigma_i^2$,
> normalized and capped to sum to one, so it is not a variation of the total learning rate
> but a relative variation of learning rates in each dimension,
> with the aim of getting better generalization. Since it is also essentially undefined as $\sigma$ approaches zero,
> and proposes no theoretical guarantees of any kind,
> we consider it unrelated to the topic of noise-adaptivity in optimization.
> The second reference (Min, Shupeng, Taihao, Huai, Xiaoyin [2024]) tackles the issue of generalization
> error in image deblurring, where the candidate image is seeked with low cross-correlation with the
> blur-induced noise, which is also a topic we consider far removed from the optimization guarantees seeked here.
> However these two works stress the idea that noise-related automatic adaptations of the learning rate
> are intuitive ideas that would be supported by the machine learning community if they came
> with trustworthy adaptive algorithms.
>
> To clarify our problematic sentence regarding "unnecessariness" of aggregation, we do not claim that averaging method never give good performance,
> but rather than many non-averaging methods give good results. In that sense, averaging is clearly
> not mandatory, and non-averaging methods have become the de-facto standard in deep learning.
> This claim is supported for instance by the GPT3 training, which uses Adam without averaging
> [Brown et al, 2020, Appendix B p43], and the
> MuZero training, which uses a momentum version without averaging [Schrittwieser et al, 2019, Ancillary file "pseudocode.py", L553], presumably identical to its predecessors AlphaGo and AlphaGoZero.
> If networks of that scale and impact work without requiring iterate averaging, then the theory of machine
> learning should try to cover these algorithms, to better explain and predict their performance.
>
> We have changed the notation of the hyperparameter in $D$-adapt from $D$ to $d_0$ for clarity,
> it is the same hyperparameter used in the original reference, which functions as a "guess" of the
> true distance $D$. The purpose of $D$-adapt is to reduce sensitivity of the algorithm to the precise
> choice of this guess, hence the "adaptivity" when this sensitivity vanishes.
>
> References:
> - Brown, Mann, Ryder, Subbiah, Kaplan, Dhariwal, Neelakantan, Shyam, Sastry, Askell, Agarwal, Herbert-Voss, Krueger,Henighan, Child, Ramesh, Ziegler, Wu,  Winter, Hesse,  Chen, Sigler, Litwin, Gray, Chess, Clark, Berner, McCandlish, Radford, Sutskever, Amodei (NeurIPS 2020) - "Language models are few-shot learners"
>
> - Schrittwieser, Antonoglou, Hubert, Simonyan, Sifre, Schmitt, Guez, Lockhart, Hassabis, Graepel, Lillicrap,  Silver (2019) - "Mastering Atari, Go, Chess and Shogi by Planning with a Learned Model", Ancillary material "pseudocode.py" (https://arxiv.org/src/1911.08265v2/anc/pseudocode.py)

---

### Official Review · Reviewer_SUNZ · 2025-06-30

**Clarity:** 2
**Significance:** 2
**Originality:** 3
**Rating:** 5
**Confidence:** 4

**Summary:**

The authors propose a quantity called “noise stability ratio” that helps them design a noise-adaptive version of SGD, Stab-SGD, that is highly robust to noise without tuning the learning rate. The noise stability ratio ranges from 0 to 1, where it is closer to 0 when the noise is high and 1 otherwise.
An efficient estimator for the noise stability ratio is provided as well, and the experiments show robust performance on synthetic problems emphasizing noise, strong convexity, and smoothness.
Specifically, Stab-SGD simultaneously performs well on strongly convex problem with low noise and smooth problems with high noise.
Experiments on ResNet-56 on CIFAR-10 show promising results as well.
The authors complement their algorithm with a last-iterate, horizon-free convergence analysis.

**Questions:**

- Figure 1 shows that Stab-SGD shows good balance, but so is the sqrt(1/t) schedule.
In Figure 2, the authors mention high sensitivity to initial leaning rate, but I don’t see that for the sqrt(1/t) schedule.
Could the authors clarify this point?

- The stability ratios in Figure 22 highly varied trajectories with a downward trend. The downward trend might seems understandable as the noise becomes more significant near the optimum, but why is it highly varied in the middle? I think it would be interesting if the authors can provide an explanation.

- Does Stab-SGD work with momentum?

- line 66: what is “not identically null”?

**Ethical Concerns:**

["NO or VERY MINOR ethics concerns only"]

**Final Justification:**

This direction of research deserves more attention, as the authors have mentioned. I believe this work is an interesting contribution, regardless of the immediate practicality of the proposed algorithm in production-grade models. It will allow other researchers and practiioners to build on it and design better noise-adaptable algorithms. I disagree with the reviewers that say that estimating the stability ratio is inefficient or that an oracle for this quantity is a strong theoretical assumption. The authors provided an efficient estimator of this quantity, and using more gradient oracle calls per step to estimate some quantity can sometimes be better than using them directly for descent (one example from the top of my head is AdaHessian).

**Limitations:**

None.

**Paper Formatting Concerns:**

None.

**Quality:**

3

**Strengths And Weaknesses:**

**Strengths**

The authors show that Stab-SGD shows “no plateau, no explosion, no saturation”, which is difficult to achieve on the the two extreme ends of problems considered (QSC and QWC) without hyperparameter tuning.

The first paragraph in Sec 4 is interesting and I agree with it. It is often good to report those hyperparameter tuning budget, particularly for papers proposing a new optimization algorithm.

Figures 3 to 6 are very informative about how parameter/schedule-free algorithms compare on two regimes: strong convexity + low noise vs. smooth + high noise. Stab-SGD does show robust and stable performance on both problems with $\\eta = 1. This is one of the strong points of this paper. (The experiments also interestingly reinforce COCOB-backprop’s position as a powerful parameter-free algorithm, especially Figure 19).

The authors provide last-iterate convergence results using a general theoretical framework and provide rates for smooth, strongly convex, and non-convex problems. I have only skimmed the proof due to time constraints. I’m not very familiar with some of the proof techniques used here, e.g., the KŁ stochastic integration lemma, but I understand that it generalizes the PŁ condition. I will review the proof in more detail later and make any amendments as necessary.

The usage of Jackknife estimator is an interesting contribution in this context. It is an efficient way to estimate the noise stability ratio.


**Weaknesses**

I think one of the weak points of this paper is the clarity of writing.
For example, the intro, while very informative, could do with smaller sentences and clearer phrasing and style. For example, an open paranthesis that extends from line 40 to line 46 is not easy to digest. Another example is the last sentence in line 48, which is not written in a natural way. The sentence “lack of [X] is supported by lack of [Y]” is difficult to digest since “lack of” is a negation, and the filler sentences make it even more difficult to parse (reminding me a little bit of Bertrand Russell’s writing).

Regarding the last paragraph in Sec 3, starting at line 186: this paragraph tries to justify the practical design of the estimation overhead control parameters. However, this control introduces three extra parameters that might not be as intuitive to the user as, say, momentum. Also, trying to choose $\\kappa$ and $\\alpha$ has an ironic parallel to choosing $\\eta_0$ and $\\alpha$ in $\\eta_0 t^{-\\alpha}$, especially given the fact that $\\eta_0 t^{-1/2}$ performs very well on both the QSC and QWC problems. The authors might argue that their parameters do not require careful tuning and can be adapted in a short horizon.
This is true, as can be seen from the experiments, but so are the parameters of $\\eta_0 t^{-\\alpha}$, as $\\alpha$ is usually one of 0, -1/2, or -1, and $\\eta_0$ is then relatively easy to tune in the short horizon. This can also be seen in the experiments. In practice, we simply choose the largest $\\eta_0$ that doesn’t blow up. The experiments in Appendix B also support this argument (except Figure 14 maybe).

In addition to the above, the edge cases of the control parameters that the authors—perhaps correctly—describe as harmless and easy to tune might not necessarly be so in real-world problems, unless the authors can demonstrate that to be the case on a wide range of problems. The introduced parameters do remind me of the warmup and exponent hyperparameters in the “schedule-free” algorithm by Defazio et al., which they tout as a minor warmup trick (but they did show competitive performances across a much wider range of real-world tasks, albeit with some learning rate tuning).

Performance on CIFAR-10 is not very promising in my opinion. It seems to require more gradient calls in order to achieve similar performance to SGD. I was hoping for the algorithm to give at least similar performances to SGD in the longer run, if not better. The algorithm also shows a sudden drop in test accuracy at some point after 2x10^4 steps and very high variance in Figure 21.

line 824, the authors mention the following: "this means that the 824 learning rate of mini-batch SGD must be re-tuned if the batch size (i.e. noise level) is altered."
This is not necessarily true. The noise introduced from mini-batching is very different from Gaussian noise. I cannot exactly recall the paper, but the learning rate can be set to be linearly proportional the batch size (or its square root). In general, some rules of thumb in practice are quite effective, so the need for full grid tuning is definitely not necessary for ERM problems.

Unfortunately, the authors did not share the code for the experiments.

**Minor**
- In line 139, is J_n the same as the numerator of the equation in Definition 3? If so, the notation should be consistent.
- I believe Sec 1 should be intro.
- Last paragraph in Sec 3: “caracteristic”, “caracteristisation” -> “characteristic” and “characteristisation”.
- Lines 144-145: What is $\\zeta$? And line 153: What is $\\alpha$? If these are the control parameters, then they should be defined before they are introduced.
- Lines 144-145: Also, could you please clarify what you mean by the last sentence?

---

> ### Author Rebuttal · Authors · 2025-07-30
>
> Thank you for your time reviewing this submission.
> We will heavily edit all introductory paragraphs to shorten sentences and improve clarity.
> We have also noted your various editing comments, and fixed the misspellings and lacking
> context on the jacknife numerator $J_n$, sample overhead $\zeta$ and multiplicative noise level $\alpha$.
> L145 now reads: "The kurtosis $\kappa = \mathbb{E}[\lVert X \rVert^4] / \mathbb{E}[\lVert X \rVert^2]^2 = 1 /\operatorname{Stab}(\lVert X \rVert^2)$ is a property of a random variable quantifying the number of samples
> needed to estimate the variance. Variables with low kurtosis have empirical estimates of the variance close
> to their true variance, whereas variables of high kurtosis require many more samples for accurate estimations
> of their variance". The "$G$ not identically null" assumption has been reworded to "$\mathbb{E}[\lVert G \rVert^2] > 0$", which is simply there to avoid a variable $G$ almost surely equal to zero, which would have undefined stability.
>
> Contrary to your first intuition, we believe the results on CIFAR-10 to be promising, for the following
> reason.
> As you have noted, typical practice in early step-size tuning is simply to pick the largest
> parameter which does not blow up, which is a behavior typical of deterministic scenarios (or negligible noise).
> This choice is supported by Figure 14 (low noise $\sigma_0^2 = 10^{-16}$) for the first $10^5$ iterations.
> However, as you have also noted two paragraphs down, neural networks also encounter at a later point
> a noisy regime, where dividing a batch size by four requires halving the learning rate.
> This noise-dependent intuition is supported by the fact that using $b \in \mathbb{N}$ samples of
> variance $\sigma^2$ leads to a batch-average of variance $\sigma^2 / b$, therefore the prescripted
> horizon-dependent learning rate $\eta_t = C \sigma^{-1} T^{-1/2}$ scales like $b^{1/2}$.
> As an aside, note that this is not supported by Fig 12-13, where SGD at $10^5$ iterations has optimal
> learning rate going from $10^{-4}$ to $10^{-3}$ (one order of magnitude) despite a variance $\sigma^2$ divided
> by $10^4$ (two orders of magnitudes in standard deviation $\sigma$), so this is really a rule of thumb
> not an analytical optimum, valid in some restricted settings and requiring more theory in others.
> The transition between these two noiseless and noisy regimes happens at $T=10^1$ in Fig 13, but at $T=10^5$
> in Fig 14 with lower noise. In neural networks, it may mean that the first 100 epochs are essentially noiseless,
> but that after that the noisy learning-rate adjustment must be performed.
> Identifying the threshold at which this switch occurs, and adjusting the learning rate of the later part,
> will surely require several training runs, say $k \in \mathbb{N}$ (with the intuition that $k \approx 10$).
> On Fig 7, without any late-horizon finetuning requiring full retrains, Alg 1 reaches the same (and even better)
> training loss in $12 \cdot 10^{4}$ iterations than SGD *with a scheduler* did in $6 \cdot 10^4$ iterations.
> This means that using this algorithm would have given better results as long as $k > 2$.
> Because we do not believe this scheduler was identified using less than two training runs,
> this means that when accounting properly for *all* training and tuning costs, and not just the single
> run presented, a noise-adaptive algorithm would have given better results at equal budget.
> It is not necessary for Stab-SGD to outperform SGD on a single isolated and already-tuned training run,
> the value of such adaptive algorithms would be in curbing the total training cost, by requiring less tuning.
>
> Note that this intuition also gives a hopeful answer to the question of new hyperparameters introduced in
> Alg 1 to estimate the stability ratio. Compared to using $\eta_0 t^\alpha$ with three full runs $\alpha \in \{0, -1/2, -1\}$ (and presumably many more to tune $\eta_0$ separately in each case), if the parameters of Alg 1
> can be roughly guessed with a rule of thumb (e.g. 10\% of budget allocated to stability estimation),
> then the adaptive algorithm will outperform a tuned version with equal tuning budget.
> We agree that the introduction of these hyperparameters in Alg 1 is unfortunate, and would not be satisfactory
> for a production-grade deployment. However, we believe that they will serve as a useful hint
> to experienced practitioners to improve upon this first proposition and eventually match Stab-SGD performance
> with little to no new hyperparameter tuning. Thus the benefits of introducing these hyperparamters in this
> submission outweigh the slight unease caused by the similarity to scheduling parameters, because the
> lower sensitivity to stability-estimating parameters will make it easier to erase their presence, be it
> by good rules of thumb or by future parameter-free alternatives.
>
> Regarding your questions,
> we leave the question of noise-adaptive last-iterate algorithms with momentum as an open problem for the moment.
> The major difficulty in extending this analysis to momentum will likely be the handling of the interaction
> between past gradient samples and current gradient, because there is not only noise in past gradients, but
> also a difference between past and current gradients that is unknown and controlled only by the smoothness.
> We also leave the question of the non-monotone stability ratio open for the time being,
> we provide this data mostly to disprove the quick intuition of a constant noise and strictly decreasing
> gradient magnitude, justifying more studies of stability estimation along trajectories.
> The sensitivity of the power schedule $\eta_t = \eta_0 t^{-1/2}$ to hyperparameter choice is best
> seen on Fig 2b, where although the same asymptotic rate is achieved (the same slope), the values
> attained at $10^8$ iterations depend on the particular choice of hyperparameter, and several are
> no better than SGD tuned at horizon $10^7$.

---

> > ### Comment · Reviewer_SUNZ · 2025-08-07
> >
> > I thank the authors for their interesting and insightful rebuttal, and for making the adjustments to improve the readability of their manuscript. Adaptability to noise is not as well studied as adaptability to other hyper-parameters (such as Lipschitzness and smoothness). I am convinced that this direction of research deserves more attention, and I believe this work is an interesting contribution, regardless of the immediate practicality of the proposed algorithm in production-grade models. I disagree with the reviewers that say that estimating the stability ratio is inefficient and uses more gradients, etc. The authors provided an efficient estimator, and using more gradient oracle calls per step to estimate some quantity can be better than using them directly for descent (quite a few interesting algorithms in the literature do that). Thus, I decided to increase my rating to "accept" on the condition that the authors pledge to release the code if the paper was accepted.

---

### Official Review · Reviewer_2Qgz · 2025-07-02

**Clarity:** 2
**Significance:** 1
**Originality:** 3
**Rating:** 2
**Confidence:** 3

**Summary:**

This paper proposes a new optimization algorithm in the smooth, stochastic setting which adapts to the noise distribution of the gradient estimates using the notion of the ``stability ratio," avoiding the need for setting the learning rate as a hyperparameter. For a random variable $X$, the stability ratio is defined as $Stab(X) = \frac{||\mathbb{E}[X]||^2}{\mathbb{E}[||X||^2]}$. Given this definition, they propose the Stab-SGD method which employs the update $\theta_{t+1} = \theta_t - \frac{1}{\beta} Stab(G_t) G_t$ where $G_t$ is the stochastic gradient estimate and the objective is $\beta$-smooth. Given oracle access to the stability ratio, they derive convergence rates for convex, strongly convex, and non-convex settings. They then propose an estimator for this stability ratio given i.i.d. samples and analyze its error. For this estimator, they propose an algorithm for implementing Stab-SGD on real data. They compare its performance relative to other learning-rate-free algorithms on synthetic quadratic loss functions and also perform a comparison to SGD for training a neural network on CIFAR-10.

**Questions:**

1. Is there any convergence analysis for the practical variant?

2. Is there a setting where Stab-SGD provably or empirically beats other methods convincingly?

3. Can the authors compare the convergence rates in Table 1 to other methods which operate under similar assumptions?

**Ethical Concerns:**

["NO or VERY MINOR ethics concerns only"]

**Final Justification:**

As I wrote in the discussion, I have the following concerns.

Firstly, there is a large gap between the algorithm for which this paper derived convergence rates (using the stability ratio) and Algorithm 1. If the stability ratio is updated in every iteration, the new theoretical statement the authors proposed in response to my comments (convergence rates with high probability if the stability ratio is approximated well enough) together with the approximation result in Lemma 1 is a step in the right direction. However, we are not able to verify the details of the proposed statement and the corresponding proof.

Moreover, the general form of Algorithm 1 does introduce new hyperparameters, which is counterproductive to the overall aim of the work.

The use and analysis of stochastic step-sizes itself is not a novelty or a major technical difficulty. It is standard practice e.g. in Reinforcement Learning.

Even taken into account the comments the authors regarding hyerparameter training budget, I still believe the numerical experiments are not convincing especially compared to the author parameter-free algorithms.

Therefore, I believe the paper has two main contributions:

1. It starts a discussion on noise-adaptive learning rates.
2. It presents an algorithm (when having access to the stability ratio) that adaptively chooses a constant learning rate for situations with asymptotically degenerate noise and a decaying learning rate for situations with noise.

I think this is a strong idea, but I am hesitant to recommend acceptance due to the limitations of the work.

**Limitations:**

The authors mention some limitations but do not discuss the practicality of the stability oracle access assumption or the downside of the additional hyperparameters introduced to estimate the stability ratio in real data settings without this oracle access.

**Quality:**

2

**Strengths And Weaknesses:**

Strengths:

The Stab-SGD successfully removes the dependence on a learning rate hyperparameter when given access to a stability oracle and only requires setting a hyperparameter relating to the smoothness of the objective. Further, the effective learning rate adapts to the noise of the gradient estimates, avoiding the need for an explicit learning rate scheduler. The theoretical convergence analysis given the stability oracle seems correct and the results are clearly presented.

Weaknesses:

The Stab-SGD method is primarily motivated as a way to eliminate additional hyperparameters such as the learning rate and learning rate schedule in a way that adapts to the gradient noise. In the theoretical setting, even given access to the stability oracle, the Stab-SGD method still requires knowledge of the smoothness parameter. Further, the practical variant that estimates the stability ratio from gradient queries requires many additional hyperparameters which set the initial learning rate, frequency of stability estimates, and number of gradient queries per stability estimate. Thus, the Stab-SGD method ultimately still requires setting many hyperparameters, which is contradictory to its original motivation.

The theoretical contributions focus on the setting where there is oracle access to the stability ratio of the stochastic gradients. This assumption is unrealistic and there is no convergence analysis of the proposed practical method which estimates the ratio from gradient queries. They provide error bounds for the finite sample stability estimate, but the impact of this error on the convergence is not analyzed.

Lastly, Stab-SGD does not seem to improve upon other schedule-free methods or SGD in either of the synthetic data or CIFAR-10 experiments, so the practical significance appears limited.

---

> ### Author Rebuttal · Authors · 2025-07-30
>
> Thank you for time reviewing this submission.
> As you noted, the purpose of this modification to the SGD algorithm is the
> *adaptivity to noise* with *last-iterate guarantees*.
> There was (until the work presented here) no guarantee that such a strong property was even possible.
> This is why general practice in deep learning is to tune both a learning rate (related to the smoothness),
> and a schedule (to account for higher noise requiring smaller steps).
>
> Therefore, we still require knowledge of the smoothness parameter, because that is not the parameter
> we aim to gain adaptivity on, we are focusing on the noise parameter.
> We also do not claim that the presented practical algorithm (Alg 1) doesn't have any hyperparameters,
> or that should
> immediately be used in place of other optimizers in production.
>
> Our theoretical contributions demonstrate that *is indeed* possible to achieve adaptivity to noise
> with last-iterate algorithms. This is a model much closer to what is sought in practical implementations
> than the typical average-iterate guarantees used in parameter-free algorithms.
> In order to push the research on adaptivity in this direction, we believe it is important
> to publish these developments, which will constitute the building blocks of a theory of last-iterate
> algorithms with good adaptive properties and fewer parameters.
>
> This new type of guarantee is desirable because noise is present in deep learning and last-iterate algorithms are the standard. Furthermore, we demonstrate feasibility of these guarantees by highlighting the central notion of stability ratios.
> We show that this variant (with stability oracles) provably outperforms other algorithms in the adaptive setting.
> For a concrete example, consider a strongly convex objective with unknown noise level.
> If the learning rate is chosen regardless of noise as $\eta_0 t^{- \alpha}$, then
> $\alpha = 0$ will yield bad performance on strictly positive noise (saturation) and $\alpha=1/2$
> will yield bad performance on null noise (stalled due to learning rates too small), while
> Stab-SGD is a unique algorithm achieving both of the rates described in Table 1,
> it is therefore an *adaptive* algorithm outperforming a *tuned* alternative in which
> $\alpha$ has to be tuned on each new problem depending on whether there is noise or not.
> To highlight why this distinction between noisy and noiseless is non-trivial,
> consider a large neural network at initialization with large batch sizes.
> Initially, all gradients will tend to point in the same direction, thus the setting is essentially
> noiseless and one should choose the largest non-exploding step-size (characteristic of deterministic settings).
> However as training
> progresses the noise will become more apparent, and thus the setting will become essentially noisy, with
> a step-size required to decrease. Because we cannot predict the threshold at which this switch will occur,
> we must perform expensive experiments with various decrease-horizons. An ideal noise-adaptive
> algorithm would eliminate the need for this expensive tuning, which is why we believe this to be a fruitful
> research direction, hence the proof of concept of a noise-adaptive last-iterate algorithm presented here.
>
> We do not claim that the stability oracle is "realistic" in any way, rather we use it as a stepping
> stone for this new adaptive setting, by showing how *any (present or future) stability-estimator*
> can be used to construct an adaptive algorithm.
>
> This new kind of guarantee with stochastic step-sizes (because $\eta_{t+1}$ depends on $X_t$ which is random)
> poses new technical challenges. We show how to overcome some of them, going up to convergence bounds
> in expectation, by leveraging new analysis techniques (KL integrations).
> However, the analysis of the practical variant (Alg 1) will pose even more new technical challenges,
> essentially because stability-estimation errors could "compound" in non-trivial ways,
> which is not something easily handled with the
> usual tools of convergence bounds in expectation, and will likely require the introduction of new tools.
> This is in our opinion all the more reason to share our progress on the technical challenges that
> we have already solved, so that others may build upon it to construct new analyses and better estimators,
> to get us closer to production-ready adaptive algorithms and guarantees closest to existing practice.

---

> > ### Comment · Reviewer_2Qgz · 2025-08-01
> >
> > Thank you for your response. I acknowledge that the primary contribution of the work is theoretical, and I appreciate the authors' candid discussion of its current practical limitations. However, the theoretical results rely on access to this stability oracle, which is a significantly stronger requirement than access to standard stochastic gradients. The impact of the work would be substantially improved if adaptivity could be achieved under more conventional assumptions, enabling fairer comparison with existing methods and potentially paving the way for practical implementation. For these reasons, I am maintaining my original score.

---

> > > ### Author Response · Authors · 2025-08-05
> > >
> > > Thank you for your comment and for acknowledging the theoretical contributions of our work. We would like to clarify our opinion on two aspects discussed:
> > >
> > > First, while our work does introduce a stability oracle, we do not believe that this constitutes significantly stronger requirement than the access to standard stochastic gradients. As described in Definition 3, the oracle can be  replaced by an estimator  constructed from gradient samples thanks to a Jackknife procedure, whose convergence with high probability is discussed in Lemma 1. Thus in the more reasonable scenario where no oracle is present,
> > > a surrogate can be constructed from commonly used and available quantities, as observed in our later experiments.
> > >
> > > Secondly, we believe that our empirical comparisons are fair and informative. For QSC and QWC problems, the fairness of comparison is ensured by presenting results as functions of a fixed hyperparameter search budget. Regarding CIFAR-10, we compare our algorithm against SGD with a scheduler, whose tuning process is non-trivial, and the cost of this tuning is rarely accounted for in practice. We present plots including both the "oracle" version (neglecting costs induced by stability-estimation) and the "raw" version (including this costly stability-estimation process), and therefore needs more iterations.
> > > The "raw" Alg 1 version and the scheduled SGD are compared with the same number
> > > of gradient queries, a comparison which we consider inherently fair and
> > > not misrepresenting the practical costs.
> > > However, as discussed with reviewer SUNZ ($\S$2), our intuition is that tuning a scheduler should necessitate $k$ runs (with $k$ of order 10 to 100, for a total cost of $k \times T$), so having a single adaptive run requiring only twice as many iterations (runtime under $2T$ observed in the ResNet experiment) as one SGD run with an already-tuned scheduler (thus a total cost of $k \times T + T \gg 2 T$ including hyperparameter-tuning budget) is to be interpreted as promising in our opinion.
> > > Indeed the aim of adaptive algorithms is to lower the hyperparameter-tuning cost,
> > > not the training cost itself, the comparison must thus be done between total training budgets.
> > >
> > > We hope this clarifies our perspective and appreciate the opportunity to respond.

---

### Comment · Area_Chair_oggL · 2025-08-05

Dear Reviewers,

Thank you again for your time and efforts in reviewing papers for NeurIPS 2025.

I am writing to remind you that **active participation in the author-reviewer discussion phase is mandatory**. According to the guidelines from the NeurIPS program chairs, reviewers are **required to engage directly with the authors in the discussion thread**, especially in response to their rebuttals.

Please note the following important policy:

- Simply reading the rebuttal or internally considering it is **not sufficient** -- reviewers must **post at least one message to the authors**, even if it is only to confirm that their concerns were resolved. If they have not been addressed, please explain why.

- **Acknowledging the rebuttal without any engagement with the authors will be considered insufficient**. I am obligated to flag such cases using the *InsufficientReview* mechanism, which may **impact future reviewing invitations and result in desk rejection of your own submissions**.

If you have not yet responded to the authors in the discussion thread, I kindly ask you to do so **as soon as possible**, and **no later than August 8, 11:59pm AoE**.

Please don't hesitate to reach out to me if you have any questions or concerns.

Best regards,

AC

---

### Decision · Program_Chairs · 2025-09-17

**Decision:**

Accept (poster)

**Comment:**

This paper proposes using the stability ratio of the stochastic gradient to adaptively adjust the stepsize in SGD to the noise level. Assuming access to a stability oracle, the authors derive last-iterate convergence guarantees for weakly convex and strongly smooth convex problems with horizon-independent stepsizes, as well as best-iterate rates for the non-convex smooth setting. They also introduce a practical variant, Stab-SGD with the Jackknife estimator, though no theoretical analysis is provided for this version.

The reviewers were divided after extensive exchanges with the authors and among themselves:

- **Reviewers hYFS and SUNZ** expressed strong support for the paper.
- **Reviewers 2Qgz and aGBz** recommended rejection, primarily due to the gap between the theoretical results and the practical implementation.

Despite this gap, I find the central idea novel and of significant interest to the community. Moreover, the demonstration of last-iterate, schedule-free convergence in stochastic settings represents a non-trivial and meaningful contribution. While the practical variant lacks theoretical backing, the idealized version of Stab-SGD comes with strong guarantees, and the authors convincingly explain how it can be approximated in practice, supported by promising empirical evidence.